# Peptides Targeting HER2-Positive Breast Cancer Cells and Applications in Tumor Imaging and Delivery of Chemotherapeutics

**DOI:** 10.3390/nano13172476

**Published:** 2023-09-01

**Authors:** Palmira Alessia Cavallaro, Marzia De Santo, Emilia Lucia Belsito, Camilla Longobucco, Manuela Curcio, Catia Morelli, Luigi Pasqua, Antonella Leggio

**Affiliations:** 1Department of Pharmacy, Health and Nutritional Sciences, University of Calabria, Via P. Bucci, 87036 Rende, Italy; alessia.cavallaro@unical.it (P.A.C.); marzia.desanto@unical.it (M.D.S.); emilialucia.belsito@unical.it (E.L.B.); camillalongobucco@gmail.com (C.L.); manuela.curcio@unical.it (M.C.); catia.morelli@unical.it (C.M.); 2Department of Environmental Engineering, University of Calabria, Via P. Bucci, 87036 Rende, Italy

**Keywords:** targeting peptides, HER2-positive breast cancer, monoclonal antibodies, nanoparticles, drug delivery, tumor imaging

## Abstract

Breast cancer represents the most common cancer type and one of the major leading causes of death in the female worldwide population. Overexpression of HER2, a transmembrane glycoprotein related to the epidermal growth factor receptor, results in a biologically and clinically aggressive breast cancer subtype. It is also the primary driver for tumor detection and progression and, in addition to being an important prognostic factor in women diagnosed with breast cancer, HER2 is a widely known therapeutic target for drug development. The aim of this review is to provide an updated overview of the main approaches for the diagnosis and treatment of HER2-positive breast cancer proposed in the literature over the past decade. We focused on the different targeting strategies involving antibodies and peptides that have been explored with their relative outcomes and current limitations that need to be improved. The review also encompasses a discussion on targeted peptides acting as probes for molecular imaging. By using different types of HER2-targeting strategies, nanotechnology promises to overcome some of the current clinical challenges by developing novel HER2-guided nanosystems suitable as powerful tools in breast cancer imaging, targeting, and therapy.

## 1. Introduction

Breast cancer (BC) is the most common cancer affecting women worldwide. An estimated 2.3 million cases of BC were diagnosed in 2020 [1] with new cases reaching 9.1 million by 2070. Worldwide, BC is the fifth leading cause of cancer-related deaths, with 685,000 deaths in 2020 [2]. Although the disease affects both genders, BC is the most frequent type of cancer in women.

BC manifests as a heterogeneous tumor encompassing a diverse array of morphological and molecular subtypes, resulting in variable biological behaviors, clinical presentations, and prognoses. BCs are classified as carcinoma in situ and invasive carcinoma, and both can be further divided into ductal and lobular if arising from the cells covering the ducts and lobules in the breast, respectively [3]. 

BC is characterized by abnormal and uncontrolled growth of malignant cells in the mammary epithelial tissue. Invasive breast cancers (IBC) include a broad spectrum of tumors differing in their clinical presentation, behavior, and morphological aspect. According to the World Health Organization, there are at least 20 major types and 18 minor subtypes of breast cancer [4]. Considering this traditional morphology-based classification, invasive breast cancer consists of about 60–75% non-specific ductal carcinoma and approximately 25% specific subtypes (e.g., invasive lobular carcinoma, tubular, mucinous A, mucinous B, and neuroendocrine) [5,6]. The molecular classification, developed with the advancement of genomic and expression profiling studies, recognizes different molecular subtypes of breast cancer based on the expression of hormone receptors. These subtypes play a crucial role in determining treatment approaches and prognosis and include the following: Luminal subtype, further divided into in Luminal A and B [7,8,9], Normal-like, HER2-enriched, and Triple Negative [7] (Figure 1). 

The underlying causes of the majority of breast cancer instances remains unknown. However, numerous risk factors for the disease have been identified, including age, genetic mutations, high hormone levels, unhealthy lifestyle, a family history of breast cancer, and exposure to environmental factors [10]. Furthermore, an increased risk in breast cancer has been associated with genetic factors, the use of hormone replacement therapy, hormonal contraception, and exposure to ionizing radiation [11]. The survival rate is very variable even though, over the years, there has been a positive trend thanks to important changes in screening methods, early diagnosis, and therapy [12]. Breast cancer treatment includes a range of approaches, such as surgery, radiation therapy, chemotherapy, hormone therapy, and more recently, immunotherapy. The main challenge is to determine the most appropriate treatment or combination of treatments tailored to the individual patient’s needs. The important adverse effects of chemotherapy due to poor selectivity towards cancer tissues over healthy ones, such as hair loss, gastrointestinal disturbances, neutropenia, and depressed immunity, have prompted the design and development of selective chemotherapeutic treatments through the use of targeting elements to deliver drugs preferentially to tumor sites.

Several studies have indicated that cancer cells exhibit different patterns of surface receptors related to their tissue type and microenvironment [13]. Therefore, a major challenge lies in identifying potential biomarkers and designing suitable targeted ligands able of precisely recognizing them. Over the past decade, breakthroughs in molecular biology and pharmacology, including advancements in the molecular subtyping of cancer, have been pivotal in understanding breast cancer, allowing for the development of innovative and smarter therapies that target cancer more specifically, thus resulting in more effective treatments. 

Among the molecular markers related to BC, HER2 has an important prognostic and predictive value for targeted treatments. HER2 has emerged as an important therapeutic target in the treatment of HER2-overexpressing (HER2+) breast cancer [14] that is considered one of the most aggressive subtypes since it is associated with an increased risk of recurrence, higher mortality in the early stage of the disease, and drug resistance [15]. In light of these premises, it seems clear that more efficient breast cancer treatment strategies, based on innovative technologies that target the HER2 molecular marker, could represent valid and effective approaches for HER2+ breast cancer therapy.

This review explores various approaches, reported in the last decade, aimed at the diagnosis and treatment of HER2-positive breast cancer, focusing on targeting strategies involving antibodies and peptides.

## 2. HER2 Receptor

The HER2 receptor (also known as HER2/Neu or ERBB2) is a 185 kDa transmembrane glycoprotein that belongs, along with HER1 (also known as EGFR or ERBB1), HER3 (ERBB3), and HER4 (ERBB4), to the epidermal growth factor receptor (EGFR; also known as HER/ERBB) family with tyrosine kinase activity. The EGFR family plays an important role in many human cancers by regulating cell growth, survival, and differentiation through multiple signal transduction pathways. 

HER2, like the other three HER receptors, is composed of three components: an extracellular ligand-binding domain, a transmembrane domain, and an intracellular cytoplasmic tyrosine kinase domain. Ligand binding to the extracellular region results in dimerization of HER receptors and activation of cytoplasmic kinase, leading to transphosphorylation of their intracellular domains and initiation of signaling pathways, primarily the mitogen-activated protein kinase (MAPK), phosphatidylinositol-4,5-bisphosphate 3-kinase (PI3K), and protein kinase C (PKC) that promote cell proliferation and invasion [16] (Figure 2).

Differently from other ERBB family members, HER2 is considered an orphan receptor since it does not bind directly to EGF-like ligands and activation results from heterodimerization, with ligand-activated EGFR or ERBB3, or from homodimerization in the case of high concentrations of the receptor (such as when overexpressed in cancer cells) [16,17]. HER2, due to its permanently open conformation, is able to dimerize with the other family members forming heterodimers with a particularly high ligand binding and signaling potency that promote cell proliferation, motility, differentiation, and survival [18]. The activation of HER2 causes un-regulated activation of the PI3K/AKT and Ras/Raf/MAPK pathways and the consequent development of various forms of cancer such as breast, lung, uterine cervix, stomach, ovary, colon, bladder, and esophagus [19,20,21,22]. Despite its involvement in the development of several cancer types, most of the studies on HER2 have been focused on breast cancer both in vitro [23] and in vivo [24]. Amplification of the HER2 gene on chromosome 17 in HER2-positive breast cancer results in the marked overexpression of HER2 on the cell surface of breast cancer, which impairs normal signaling function. HER2 gene amplification is associated with shorter overall survival and is considered an important predictor of both overall survival (*p* < 0.001) and time to relapse (*p* < 0.0001) in breast cancer [25,26].

Furthermore, it has been found that in HER2-positive breast cancers, also expressing steroid hormone receptors for estrogen, progesterone, or both, the content of steroid hormone receptors is lower than in HER2-negative, hormone-receptor-positive tumors. As a result, HER2-positive breast cancer often exhibits resistance to tamoxifen, a common hormone therapy used in hormone-receptor-positive breast cancers [27].

HER2 is overexpressed in approximately 30% of invasive breast cancer cases, with adverse prognostic implications associated with high-grade tumors, high rates of cell proliferation, increased incidence in brain metastasis, and drug resistance [27,28,29,30]. Therefore, HER2 overexpression can serve as both a marker of aggressive disease and an important target for the diagnosis and treatment of human breast cancer [26,31].

## 3. HER2-Positive Breast Cancer Targeted Therapies

Currently, the major challenge in cancer therapy lies in effectively delivering chemotherapy drugs, contrast agents, and radionuclides to tumor tissues. To overcome the critical limitations of conventional cancer therapy, targeted therapies have emerged as an attractive solution. These therapies are designed to specifically target cancer cells without affecting healthy tissues, thus offering a more focused and precise approach in the fight against cancer.

In breast cancer, the overexpression of the HER2 receptor makes it a reliable biomarker and a successful therapeutic target. Several strategies have been developed to target HER2, using various targeting molecules including monoclonal antibodies and tyrosine kinase inhibitors, antibody–drug conjugates, small molecules, and peptides.

### 3.1. HER2-Targeting Monoclonal Antibodies

HER2-targeted therapies have revolutionized the strategy for HER2-positive breast cancer treatment, both in metastatic and early stage disease. HER2 monoclonal antibodies (mAbs) represent highly specific therapies with moderate toxicity, which significantly improve the clinical outcome for breast cancer patients. However, some patients experienced relapse due to an acquired therapeutic resistance to mAbs. In response, the development of antibody–drug conjugates (ADCs) has emerged as a promising class of therapeutics that combines the specificity of monoclonal antibodies with the antitumor activity of cytotoxic agents, known as payloads linked to mAbs by means molecular linkers [32], thereby maximizing the antitumoral activity [33]. 

Trastuzumab, also known as Herceptin, is a recombinant humanized monoclonal antibody that binds domain IV of the extracellular segment of the HER2 tyrosine kinase receptor [34,35]. The antitumor efficacy of trastuzumab, in combination with established therapeutic agents, has been extensively investigated through in vitro and in vivo studies on SKBR3, MCF7, and BT-474 cell lines. In 1998, the U.S. Food and drug Administration (FDA) approved Trastuzumab as one of the first available targeted chemotherapies. It received approval as an adjuvant therapy when used in combination with anthracycline- or a taxane-based chemotherapy for HER2-positive breast cancer treatment, and in metastatic HER2-positive breast cancer as a monotherapy or in combination with paclitaxel [36]. It selectively exerts anticancer effects in HER2-positive breast cancer patients [37]. Trastuzumab, when used in combination with standard chemotherapy, has shown significantly improved response rates compared to chemotherapy alone [38,39]. As a result, treatment regimens that include trastuzumab have become the standard of care for patients with HER2-overexpressing breast cancer [40]. Despite the fact that treatment with trastuzumab is considered clinically very effective in HER2-overexpressing breast cancer, its mechanism of action is not yet well-understood. In vivo breast cancer models and clinical trials indicated that antibody-dependent cell-mediated cytotoxicity (ADCC) is the most supported mechanism of action [41,42].

However, several clinical studies showed a 48% clinical benefit rate in patients treated with trastuzumab monotherapy, indicating that a significant number of HER2-amplified metastatic breast cancers do not respond favorably to monotherapy [43]. Furthermore, another important concern is the development of acquired resistance, which frequently occurs and restricts the effectiveness of trastuzumab-based therapy to the duration of the treatment [44,45]. Overall, due to breast cancer heterogeneity and the accumulation of intracellular alterations, multiple mechanisms are responsible for resistance occurrence [46]. Both innate and acquired resistance, in trastuzumab treatments, are progressively becoming a major clinical issue [47,48,49]; therefore, further and new approaches are continuously explored in order to develop more specific and effective combination therapies.

In 2013, the conjugated emtansine-trastuzumab (T-DM1) was globally approved for treating trastuzumab-resistant patients [50,51,52]. T-DM1 is a therapeutic agent composed of trastuzumab conjugated to DM1 (Figure 3), a potent microtubule-disrupting drug. This conjugated form efficiently inhibits the growth of cells and tumors that are refractory to trastuzumab [53]. It acts by combining the targeted internalization of the cytotoxic molecule (DM1) with the inherent antitumor properties of trastuzumab. Data from pivotal trials such as EMILIA and TH3RESA indicated T-DM1 as a second-line treatment for metastatic conditions [54,55,56]. 

Trastuzumab-deruxtecan (T-DXd) was the second HER2 ADC to obtain FDA approval in 2019, for treating advanced HER2-positive disease in patients who have previously received at least two anti-HER2-based regiments in the metastatic setting [57]. 

T-DXd is a unique combination of trastuzumab and a topoisomerase-I-inhibitor, DX-8951 (exatecan), linked by a tetrapeptide-based linker that is highly stable in plasma thus reducing the potential for systemic toxicity. Once T-DXd has reached the tumor tissue and is internalized by HER2-positive cells [58], the linker is selectively cleaved by lysosomal cathepsins, releasing the cytotoxic payload and enabling targeted anticancer effects. Compared to T-DM1, T-DXd presents a higher drug to antibody ratio and incorporates a cleavable linker, which likely contributes to its enhanced and more efficient anticancer activity [59,60].

The Phase 2 TUXEDO-1 trial, conducted as an open-label single-arm study, was undertaken to explore the therapeutic efficacy of T-DXd treatment in patients afflicted with active brain metastases originating from HER-positive breast cancer. The study also served as a demonstration that antibody–drug conjugates (ADCs) can effectively exhibit activity within the intracranial compartment. The results of the TUXEDO-1 study showed the clinically pertinent efficacy of ADC trastuzumab deruxtecan with comparable response rates observed both within the intracranial and extracranial domains, in a population that had previously received pretreatment. Additionally, the data concerning progression-free survival (PFS) suggest an important extension in disease control despite the presence of brain metastases [61]. 

Pertuzumab (rhuMAb-2C4) is a humanized mAb, based on human immunoglobulin G1 (IgG1) [51], belonging to a new class of drugs known as dimerization inhibitors. It binds the human epidermal growth factor receptor 2 (HER2), inhibiting the heterodimerization of HER2 with other HER receptors. Pertuzumab, which has a mechanism of action that is complementary to that of trastuzumab, is able to circumvent different mechanisms of resistance to trastuzumab, showing promising efficacy when combined with trastuzumab in several treatment settings. This mAb targets a different extracellular subdomain of HER2 (subdomain II) than Trastuzumab (subdomain IV) and, when it is administered together with trastuzumab and taxanes, shows a strong survival benefit for patients with metastatic HER2+ breast cancer [62]. In 2012, recognizing its potential, the FDA granted approval for Pertuzumab use in combination with trastuzumab and docetaxel in patients with HER2-positive metastatic breast cancer (MBC) who had not previously received anti-HER2 therapy or chemotherapy for metastatic disease. 

In April 2013, the FDA granted accelerated approval for Pertuzumab as a neoadjuvant treatment. This was the first application for the neoadjuvant treatment of breast cancer [63]. The efficacy of the combined treatment with trastuzumab and Pertuzumab was further confirmed through in vitro studies conducted on BT474 breast cancer cells, as well as in vivo studies using Calu-3 and KPL-4 xenograft tumor models in mice. These studies showed that the combined trastuzumab and Pertuzumab treatment is additive and gives results that are superior to the monotherapy resulting in enhanced antitumor activity [51]. 

Margetuximab is a chimeric anti-HER2 monoclonal antibody, based on the murine precursor of trastuzumab with a modified Fc domain that results in an increased binding capacity to CD16A (FcγRIIIA) and a reduced binding capacity to inhibitory FcγRIIB (CD32B) compared to trastuzumab. Margetuximab not only retains the antiproliferative effects observed in trastuzumab, but also possesses the ability to enhance the immune response [64,65]. Results from an initial Phase 1 human clinical trial of Margetuximab in patients with HER2-expressing tumors showed that Margetuximab monotherapy is well tolerated and has promising activity [66]. In the phase 3 SOPHIA clinical trial, the clinical efficacy of Margetuximab was investigated in comparison to trastuzumab, both in combination with chemotherapy, for patients with metastatic HER2-positive breast cancer. This study showed a highly favorable benefit–risk profile for Margetuximab when administered alongside chemotherapy [67,68]. The SOPHIA study’s findings led to the US FDA approval of Margetuximab in 2020. This approval was based on the evidence of improved progression-free survival (PFS) observed in patients treated with Margetuximab in comparison to those receiving trastuzumab plus chemotherapy.

Approximately fifty percent of breast tumors designated as HER2-negative show an expression of HER2-low-positivity. Dieci and coworkers [69] conducted an extensive investigation into the dynamics of HER2-low-positive breast cancer, exploring its progression from the primary tumor stage to the residual disease state in a large cohort of patients undergoing neoadjuvant treatment. By analysing the molecular alterations and histological transformations that occur in HER2-low-positive breast cancer, the study aims to provide a deeper understanding of its clinical behavior and therapeutic implications. Through the tracing of this evolution from primary tumor to residual disease, the research offers valuable insights to enhance the precision of treatment strategies and improve patient outcomes in this specific breast cancer subset. 

In the context of clinical trials involving novel anti-HER2 antibody–drug conjugates (ADC), the presence of HER2-low-positive expression on residual disease (RD) holds the potential to steer tailored adjuvant treatments for patients at high risk.

### 3.2. HER2 Receptor-Targeting Peptides

Monoclonal antibodies are stable molecules known for their high target selectivity. However, their production costs can be significant, making them relatively expensive. Furthermore, they have properties that may limit their use in some cases, such as large size, reduced tissue penetration due to steric hindrance, and poor efficacy for targeted applications to the brain, since in general they cannot cross the blood–brain barrier [70].

Small peptides have emerged as a highly attractive alternative to mAbs as targeting agents for human cancers since they offer a unique combination of advantageous properties that make them promising candidates for therapeutic applications. Combining the advantages of small molecules, including oral bioavailability and high membrane permeability, with those of proteins, such as target specificity, high potency of action, and relatively few off-target side effects, small peptides present a compelling option [71,72]. 

Over the last few decades, tumor-targeting peptides (TTPs) have become appealing tools in targeted cancer therapy. Due to their lower molecular weights, peptides are much easier and cost-effective to produce compared to mAbs. The unique properties of TTPs, such as their low immunogenicity, easy modification, high tumor penetration, tumor-homing capacity, and low bone marrow accumulation, highlight their potential as promising building blocks for the development of targeted drug delivery systems as well as probes for molecular imaging for cancer treatment [70,73,74]. To this aim, the solid-phase peptide synthesis method [75] emerges as a highly efficient technique, demanding only moderate reaction conditions. This method allows for the straightforward production of peptides with well-defined functions, showing specific bioactivities such as receptor recognition, cellular pathway activation, or inhibition. For cancer treatment, peptides can be used directly as drugs or as delivery vehicles of imaging agents and cytotoxic compounds to tumor tissues. In this case, cytotoxic drugs are conjugated to specific tumor-homing peptides with the aim of decreasing the off-target toxicity of the compound and delivering higher concentrations to the target. An important feature of drug-containing conjugates is that the antineoplastic agent is guided and easily released from the conjugate at the target site in order to achieve effective inhibition of tumor growth. To this aim, the insertion of pH-sensitive or enzymatically cleavable linkers between the drug and the peptide plays a key role in the circulation time of the conjugate and drug release at the target site [76,77]. Over the past decade, there has been significant discovery and reporting of various HER2-targeting peptides. 

Qiaojun Fang et al., conducted an investigation based on the interactions between HER2 and its affibody Z(HER2:342) [78] leading to the identification of 2 novel peptides containing 27 amino acid residues, pep27 (NKFNKGMRGYWGALGGGNGKRGIRGYD), and pep27-24M (NKFNKGMRGYWGALGGGNGKRGIMGYD). Both peptides showed high affinity and specificity against HER2. ZHER2, an affibody composed of 58 amino acids, adopts a stable three-helix structure consisting of helix1 (Asn6-Leu18), helix2 (Asn24-Asp36), and helix3 (Ala42-Gln55) (Figure 4). The crystal structure of the HER2/ZHER2 complex (PDB ID: 3MZW) revealed that ZHER2 binds to HER-2 at distinct sites, showing alternative properties compared to HER2 antibodies (Trastuzumab and Pertuzumab).

ZHER2 is useful for the development of imaging agents or as a vector to direct drugs towards the HER2 target. The interactions between HER2 and the two short peptides pep27 and pep27-24M were studied by combining molecular dynamics (MD) modeling with MM/GBSA binding free energy calculations and binding free energy decomposition analyses. This study showed that both peptides, pep27 and pep27-24M, are able to bind with high affinity to the extracellular domain of the HER2 protein with dissociation constants (KD) of 346 and 293 nmol/L, respectively. In vivo and ex vivo fluorescence imaging of tumors targeted by pep27 and pep27-24M showed that both peptides have strong affinity and specificity for HER2-positive tumors. In addition, both peptides showed no significant cytotoxicity, measured by MTT assay, even at high concentrations (50–100 μM) against SKBR3 (HER2 high expression) breast cancer cells [79,80].

In a subsequent work, the same research group identified four additional peptide sequences: P51, P25, P47, and P40. These peptides displayed high affinity against the HER2 protein, making them promising tools for applications in HER2-positive breast cancer imaging and targeted drug delivery [14]. The screening process of these peptides targeting the HER2 receptor involved the design of an OBOC (computational-aided one-bead-one-compound) peptide library combined with in situ single-bead sequencing microarray methods. As a result, 72 peptides were identified, and 4 of them in particular, 2 peptides with the lowest binding free energy (P51: CDTFPYLGWWNPNEYRY and P25: CKTIYYLGYYNPNEYRY) and 2 with the highest (P47: CDYIPYLAYYNPNTYFQ and P40: CKKIPPLGWWNPNTWRY), were synthesized and analysed by the SPRi (Surface Plasmon Resonance Imaging) method in order to determine their binding affinity toward HER2. The results of the SPRi analysis indicated that all four peptides have high binding affinity to the HER2 protein, with the best affinity of P51 with a KD value as low as 18.6 nmol/L. The high affinity and specificity of peptides P51 and P25 to HER2-positive cells were confirmed in vitro by flow cytometry analysis on the HER2-positive human breast cancer cell line SKBR3 and the HER2-negative human embryonic kidney cell line 293A treated with fluorescein isothiocyanate (FITC) labeled peptides P51 and P25. Furthermore, confocal fluorescence imaging analysis on HER2-positive SKBR3 cells and HER1-positive but HER2-negative 468 cells provided additional evidence of the peptides’ selective targeting. In addition, in vivo and ex vivo imaging were consistent with the in vitro findings, confirming the ability of P51 and P25 peptides to effectively target HER2-positive tumors.

In this study, the authors observed an enhanced cytotoxicity against HER2-positive cells when using doxorubicin (DOX)-loaded liposome nanoparticles that were modified by P51 and P25 peptides. This finding indicates that these peptides can be successfully used in targeted drug delivery for cancer treatment. Confocal microscopy images of SKBR3 cells treated with DOX-loaded liposome with peptides (P51-LS-DOX and P25-LS-DOX) or without peptides (LS-DOX) showed that the targeting effect of peptides occurs at the early stage of binding (within the first 5 min). In addition, an MTT Cell Viability Assay of SKBR3, treated with P51 and P25-modified DOX-loaded liposomes, demonstrated that the cytotoxicity of targeted liposomes, P25-LS-DOX and P51-LS-DOX, was significantly higher than that of non-targeted liposomes when the concentration of DOX exceeded 50 μg/mL. This suggests that the endocytosis of peptide-modified liposomes most likely occurs via receptor-mediated and non-specific uptake, while LS-DOX liposomes enter the cells only through a non-specific pathway.

In a recent work, Hai Qian et al. [81], investigating the binding mode of the Trastuzumab antibody with HER2 protein, designed and synthesized a cyclic peptide, Cyclo-GCGPep1, with good affinity towards HER2 which was able to specifically target camptothecin to HER2-positive cells via a peptide–drug conjugate (PDC). The authors first identified a lead sequence, Leadpep, containing 10 residues (RIYPTNGYTR) that is crucial for the interaction between trastuzumab and HER2 protein. Next, the researchers systematically replaced each residue of the Leadpep sequence with other natural amino acids and, through in silico analysis, evaluated the effect on binding affinity by calculating the mutation energy. As a result, they identified five peptides with triple mutations that exhibited the lowest mutation energy and selected them as potential candidates to target the HER2 protein (Table 1). Among these peptides, Pep1 showed the best binding affinity with HER2 protein, as confirmed by surface plasmon resonance (SPR) experiments, which determined a Kd (dissociation constant) value of 7.595 μM.

In order to introduce conformational restrictions in the peptide and improve its stability, the Pep1 sequence was cyclized incorporating the Gly-Cys-Gly (GCG) sequence into the Pep1 chain, which was required for the subsequent conjugation of the drug. The obtained cyclic peptide, Cyclo-GCGPep1, showed a smaller Kd (2.555 μM) compared to the linear Pep1, indicating a higher binding affinity which could therefore result in a better HER2-targeting ability. Therefore, Cyclo-GCGPep1 was used to develop peptide–camptothecin conjugates by linking camptothecin to the cysteine residue of Cyclo-GCGPep1 via disulphide bonds. Among the synthesized conjugates, Conjugate 1 (Figure 5) showed the most potent antiproliferative activity against SK-BR-3 and NCI-N87 cells. Additionally, it demonstrated excellent specific delivery capacity to HER2-positive cells, as well as better penetration compared to camptothecin used alone. These promising results suggest that Conjugate 1 represents a promising therapeutic option for the treatment of HER2-positive cancer.

Another group [70] used the monoclonal antibody Trastuzumab (Fab) to design peptide ligands specifically targeting the HER2 receptor by studying, through computational approaches, the interactions between these ligands and HER2-DIVMP (HER2-domain IV-mimicking peptide), a specific model system of HER2 domain IV [82]. Binding experiments between the selected ligands and the receptor fragment HER2-DIVMP were performed by using the receptor fragment approach by means of two techniques: the fluorescence spectroscopy and surface plasmon resonance (SPR) [83]. Among the peptides studied, the low molecular weight peptide A9 (Ac-Q27-D28-V29-N30-T31-A32-V33-A34-W35-NH_2_), selected from the Fab heavy chain of trastuzumab, showed a dissociation constant in the low nanomolar range; moreover, further structural investigations (computational method and NMR validation) on the molecular interaction between A9 and the receptor model confirmed the high binding affinity of A9 towards HER2-DIVMP [84]. Hence, the A9 peptide ligand was chosen as a suitable candidate for the development of HER2-specific radioactive probes [85]. For this purpose, the N-terminus of A9 was conjugated, via an amide bond, with the acyclic chelator DTPA (diethylenetriaminepentaacetic acid) and subsequently radiolabeled with 111In; thus, the affinity of the 111In-DTPA-A9 conjugate to HER2-positive human breast cancer BT474 cells has been investigated. The radioactive probe showed a high interaction target-specific (KD = 4.9 nM) to HER2-overexpressing cancer cells. Furthermore, biodistribution data of 111In-DTPA-A9 in normal mice showed that it does not bind healthy organs and tissues to any significant measure. The collected data confirmed that the A9 peptide is able to target the HER2 receptor with high affinity, paving the way for the use of this peptide as a promising probe for molecular imaging diagnostics and active targeting of anticancer drugs.

To address the problem of drug resistance, Koji Kawakami et al. developed a novel molecular-targeted drug, the so-called “hybrid peptide”, consisting of a target-binding peptide and a lytic peptide rich in cationic amino acid residues that disrupt the cell membrane inducing cancer cell death through membrane lysis [86]. In 2013 [87], the same group, relying on this hybrid system, developed a new HER2-targeting peptide, called HER2-lytic hybrid peptide, by combining a previously identified HER2-binding peptide sequence [88] with a lytic peptide, and investigated its cytotoxic activity in vitro and in vivo. The HER2-lytic hybrid peptide consists of two functional domains, the HER2-targeting peptide KCCYSL and the lytic peptide KL**L**LK**L**L**KK**LLK**L**LKKK (bold and underlined letters indicate D-amino acids) linked together to form the hybrid peptide KCCYSLGGGKL**L**LK**L**L**KK**LLK**L**LKKK capable of binding to HER2 and to provoke cell death by lysis. The HER2-lytic hybrid peptide was tested in 13 cell lines (1 ovarian cancer, 10 breast cancers, and 2 normal). The results of this study suggested that it specifically binds to HER2 and selectively kills HER2-overexpressing cancer cells, including trastuzumab- and/or lapatinib-resistant MDA-MB-453 and MDA-MB-361 cells, but not normal cells, and that it inhibits HER2 signaling. The antitumor activity of HER2-lytic peptide was assessed in mouse BT-474 and MDA-MB-453 xenograft tumor models. The results of the in vivo study showed that HER2-lytic hybrid peptide significantly inhibited tumor progression at the dose of 3 mg/kg.

Recently, Hosseinimehr S.J. and colleagues [89] developed a novel peptide-based 68Ga-PET radiotracer (68Ga-DOTA-(Ser)3-LTVSPWY) for HER-2 detection in cancer. Over the years, different targeted ligands such as antibodies, nanobodies, small scaffold proteins and peptides have been combined with various radioisotopes (18F, 124I, 44Sc, 89Zr, 64Cu, and 68Ga) in order to obtain PET radiotracers for imaging HER2-positive tumors [90,91]. Small peptides have many advantageous characteristics for the development of imaging agents over macromolecules such as protein and antibodies. These benefits include higher tissue penetration and faster circulation time in the blood resulting in quicker imaging time when radionuclide peptides are used for imaging of HER2-expressing tumors [92]. 

The LTVSPWY peptide sequence, identified using phage display technology, [93] or peptides with LTVSPWY incorporated into the sequence, have shown specific binding to HER2-overexpressing cancer cells. Hosseinimehr and colleagues labeled the core peptide LTVSPWY with the addition of the bifunctional chelator HYNIC (hydrazinonicotinamide) for imaging studies of HER2-positive tumors [94,95]. They also used 99mTc (Technetium-99m) to label two peptides with an LTVSPWY sequence conjugated to cysteine-based chelators (CGGG and CSSS) by obtaining the 99mTc-CGGG-LTVSPWY and 99mTc-CSSS-LTVSPWY peptides that showed significantly higher binding to SKOV-3 (HER2+) cells compared with A549 and MCF-7 (HER2-) cells [96]. In a recent study [89], the same team developed a HER2-binding peptide (DOTA-(Ser)3-LTVSPWY) (Figure 6) labeled with 68Ga (Gallium-68) and evaluated its properties in vitro and in vivo.

The imaging agent, 68Ga-DOTA-(Ser)3-LTVSPWY, consists of the HER2-targeted peptide, LTVSPWY, conjugated, through a spacer of three serine amino acid residues, to the bifunctional chelator DOTA labeled with 68Ga-radionuclide. 68Ga-DOTA-(Ser)3-LTVSPWY was evaluated in vitro in the HER2-positive human ovarian cancer cell line SKOV-3a. Subsequent in vivo studies encompassed biodistribution and imaging analyses in mice harboring xenografted SKOV-3 tumors. The developed PET (positron emission tomography) imaging probe exhibited good stability and specific binding to the HER2 receptor in the low-nanomolar range; furthermore, 68Ga-DOTA-(Ser)3-LTVSPWY revealed specific tumor accumulation and high-contrast imaging in HER2-expressing xenografts. 

The KCCYSL peptide, first discovered by Quinn et al. [88], and peptides containing the KCCYSL sequence, also specifically target the HER2 receptor. These peptides, radiolabeled with 111In and 64Cu, have been assessed in numerous studies in both in vitro and in vivo [97,98,99]. 

Gábor Mező et al. [100] chose this peptide as the starting sequence to obtain new modified peptides able to bind HER2-overexpressing breast cancer cells with high affinity and specificity. They combined the modified KCCYSL sequence with the GYYNPN peptide, taken from the computational-aided one-bead-one-compound (OBOC) peptide library [14], to yield peptide analogues targeting the extracellular region of HER2 and compared their binding to HER2-expressing cells. A set of hexa- and 12-mer peptide analogues were synthesized by solid-phase peptide synthesis (Table 2). In the last step, the fluorescent dye 5(6)-carboxyfluorescein (CF) was conjugated to the N-terminal residue of the peptide chain (Table 2).

Their extracellular localization and specificity were measured and confirmed by flow cytometry and confocal microscopy using HER2-overexpressing MDAMB-453 breast cancer cells. Changes in the amino acid sequence of the hexapeptide KCCYSL were investigated based on the fluorescence intensity measured after incubation of cells with CF-labeled peptides. The best analogues P(SC) and P(AA), showing the highest uptake in this set, were selected for the design of combined peptides. The most promising combined peptide cP(AA)_P(YY) showed ten times higher fluorescence intensity values than the hexapeptide P(AA) in the cellular uptake study. Furthermore, the peptides were detected at the membrane of MDA-MB-453 cells demonstrating their binding to the extracellular domain of HER2. The specificity of the peptides that showed the highest cellular uptake, cP(AA)_P(YY) and cP(SC)_P(YY), was monitored pre-incubating the cells with unlabeled peptides before adding CF-labeled derivatives. The flow cytometry results demonstrated that the fluorescent signal was decreased, suggesting that unlabeled peptides bind to the specific receptors on the cell surface inhibiting the binding of the CF-labeled peptides. A reversed version (cP(YY)_(P(AA)) of the best combined peptide cP(AA)_P(YY) was also analysed by flow cytometric and confocal microscopic analysis. The obtained results confirmed the importance of the order of the two peptides since the reversed peptide cP(YY)_(P(AA) binds to cells to a much lower extent. By modifying and combining two sets of known HER2-binding peptides, a 12-mer peptide was developed that binds to HER2-overexpressing cells with high affinity and specificity for use in cancer diagnostics and drug targeting.

Peptide-based therapies represent promising avenues for cancer diagnosis and therapy. Nevertheless, it is essential to acknowledge the existing limitations and challenges. Peptides, while offering immense potential, can encounter issues such as stability concerns, susceptibility to enzymatic degradation, and poor membrane permeability due to their size and hydrophobic nature. Their short circulation half-life within plasma and rapid in vivo clearance further impact their efficacy [72,101,102].

These intrinsic properties often limit their administration to the intravenous route, affecting patient compliance [103]. To overcome these limitations, strategies involving peptide modification and conjugation with stabilizing agents emerge as viable solutions to enhance peptide stability. Addressing the challenge of peptide biodistribution is equally critical, as their structural flexibility can lead to off-target interactions and trigger unwanted side effects [104]. 

Therefore, different chemical strategies have been explored to refine peptide design, focusing on increasing secondary structure stability and improving overall bioavailability. These techniques include amino acids substitution (D- or β-amino acids), cyclization, N-methylation [105,106,107,108,109], PEGylation, lipidation, and stapling to stabilize α-helices [110,111,112].

An additional aspect deserving attention is the potential for synthetic peptides to trigger immunogenic responses within the body [113,114]. Such immune reactions can compromise therapeutic efficacy and lead to potential adverse effects, necessitating careful modification to mitigate these challenges.

Furthermore, it is pertinent to highlight the economic aspect, as the complex synthesis and purification processes involved in creating these peptides can be laborious, time-consuming, and expensive [115,116]. Ensuring reproducibility, scalability, and product quality is crucial for successful clinical translation.

The risk of target cells developing resistance or adaptations, especially in the context of cancer treatment, underscores the importance of combining therapies or developing innovative targeting approaches. Additionally, challenges related to tissue penetration and intracellular delivery may be mitigated through the use of innovative delivery systems such as nanoparticles.

## 4. Applications of Peptide-Conjugated Nanoparticles in Breast Cancer Treatment and Imaging

Nanoparticles own unique physico-chemical features, such as small size, large surface-to-volume ratio for drug loading, easy surface functionalization, and surface binding sites for various ligands targeting specific receptors overexpressed by cancer cells that can offer the best solution for cancer diagnosis and therapy [117]. Their use as drug delivery systems (DDSs) helps to prevent premature drug release before reaching the destination and unwanted side effects, to reduce the frequency of drug administration resulting in increased patient compliance, and to enhance drug efficacy through targeted delivery of the drug to target cells or organs [118,119,120,121,122]. Striking the right balance among the functionalities of the nanosystem is crucial to ensure optimal performance as highlighted in recent research [123]. 

The singular optical, magnetic, and chemical properties of the nanostructured materials allow the development of imaging probes with high sensibility, controlled biodistribution, and better spatial–temporal resolution [124] resulting in clinical benefits including early detection and real time monitoring of disease progression and the translation into personalized medicine. Different types of nanocarriers, such as liposomes, micelles, polymers, quantum dots, and inorganic nanoparticles have been widely used in cancer therapy and diagnosis. Among inorganic nanoparticles, mesoporous silica-based nanoparticles represent a promising and innovative tool for cancer-related applications and more generally in bionanotechnology [125,126,127].

In this scenario, peptide–nanoparticle conjugates have emerged as versatile platforms for biomedical applications, including cancer diagnosis and treatment [128] (Figure 7).

### 4.1. Application of HER2-Targeted Peptides in Drug Delivery

Nanoparticle-based drug delivery is one of the successful approaches for reducing drug off-target effects and increasing tumor-site drug concentration. Furthermore, the conjugation on the nanocarrier surface of peptide ligands targeting specific receptors overexpressed on cancer cells allows selectively in delivering the drug to the target site. Peptides acting as tumor-targeting ligands show a high potential for the treatment of cancer including breast cancer and provide the opportunity for targeted drug delivery [129]. The highly specific targeting ability of peptides is crucial for effective cancer treatment with limited side effects.

To overcome the bioavailability issues of the chemotherapeutic agent capecitabine (CAP), Ganesh et al. [130] developed CAP-loaded liposomes (CAP-LPs). CAP-LPs were also conjugated to a tumor-homing peptide (THP) containing the WNLPWYYSVSPTC sequence to achieve specific drug delivery to HER2+ breast cancer cells. The developed formulation THP-CAP-LP was optimized by using central composite design (CCD). Preliminary in vitro tests on HER2+ MDA-MB453 cells demonstrated that the nanocarrier surface modification with THP crucially increased the CAP uptake promoting targeted therapy. Interestingly, THP-CAP-LPs also exhibited a higher cytotoxic activity than CAP and unmodified CAP-LPs, which could be ascribed to the ability of the THP-modified system to interfere with apoptotic signaling pathways.

The biophysical and chemical properties of peptide ligands are critical factors in determining the therapeutic potential of peptide-functionalized nanoparticles for targeted cancer treatment and tracking [131,132,133,134].

A study by Bilgicer and co-workers [135] focused on developing a method to improve the uptake of peptide-targeted nanoparticles by cells. The authors found that by increasing the hydrophilicity of the targeting peptide and optimizing the length of the linker between the peptide and the nanoparticle surface, they were able to enhance cellular uptake of peptide-targeted liposomes and micelles. To achieve this goal, the researchers synthesized nanoparticles that were functionalized with a targeted peptide that specifically recognizes a receptor expressed on the surface of the target cells. Functionalization of the nanoparticles with this targeted peptide improved their selectivity towards the target cells. Subsequently, various versions of peptide-targeted liposomes and micelles with different hydrophilic properties were synthesized. Peptide-targeted liposomes were prepared using peptide–lipid conjugates that consist of a receptor-specific peptide, an ethylene glycol (EG), a spacer (EG2), a oligolysine chain (KN; N is the number of repeat lysine (K) units) to increase the hydrophilicity of the targeting peptide, an EG peptide linker, and two fatty acid chains with hydrophobic properties for the insertion into the lipid bilayer of liposomes (Figure 8).

A series of peptide (K_N_)–EG_linker_–lipid conjugates with different hydrophilicity and linker lengths were synthesized and attached to the surface of nanoparticles. Then, the cellular uptake of the resulting nanoparticles was tested in vitro using a variety of cancer cell lines expressing different receptors, including the HER2 receptor. The cyclic peptide sequence, YCDGFYACYMDV called HER2pep (Figure 9), that specifically targets the HER2 receptor [136], was tested on HER2+ SK-BR-3 breast cancer cells to explore how modifications to HER2pep, specifically increasing its hydrophilicity and optimizing the length of the ethylene glycol peptide linker, could enhance the cellular uptake of HER2pep-targeted nanoparticles.

The study demonstrated a significant enhancement of cellular uptake in SKBR-3 cells when both liposomal and micellar nanoparticles incorporated three lysine residues and an EG18 peptide linker. Interestingly, the EG18 linker is the optimal spacing needed to allow HER2pep to adopt an appropriate conformation to bind to the target receptor and allow for multivalent interactions. Overall, this work highlights the potential of peptide-targeted nanoparticles for targeted drug delivery to cancer cells and suggests that modifying the hydrophilicity of peptides and optimizing linker length can further enhance the efficacy of these systems.

In a later work, built upon the previous paper, the same authors [137] extended their investigation to optimize the design of liposomes for targeted drug delivery by modifying the peptide density on the liposome surface in addition to the hydrophilicity and linker length. Four HER2 antagonist peptides, HERP5, HRAP, KAAYSL, and AHNP, (Figure 10) designed to bind to the HER2 extracellular domain [136,137,138,139] were synthesized with varying structure, hydrophilicity, binding affinity, and peptide density to investigate their effect on the targeting efficiency of the peptide–liposome system. The liposomes composed of a mixture of phosphatidylcholine, cholesterol, and PEG-DSPE and peptide (KN)–EGlinker–lipid conjugate were prepared using a thin-film hydration method and then conjugated with the obtained peptides.

The peptides were conjugated to the surface of the liposomes using different linker lengths and peptide densities, and the resulting peptide-targeted liposomes were evaluated for the cellular uptake and binding affinity to both HER2-expressing cell lines BT-474 and SK-BR-3. The study found an important improvement in cellular uptake with three lysine residues (HERP5(K3), HRAP(K3), KAAYSL(K3), and AHNP(K3)) and a maximum activity using EG6 linker for HERP5, HRAP, and KAAYSL, and EG18 linker for AHNP. Furthermore, the authors observed that for all peptide-targeted liposomes, the uptake efficiency of the peptide-targeted liposomes by HER2-positive cancer cells reached a maximum and plateaued at around 2% peptide density. The peptides KAAYSL and HRAP, which are more hydrophilic, exhibited higher availability at lower densities compared to the peptides HERP5 and AHNP, which are more hydrophobic. This is likely due to their decreased interaction with the lipid bilayer, and instead, a greater affinity for the aqueous exterior of the liposomes. 

Chemotherapeutic drugs usually display several severe side effects, and their mitigations and maintenance of therapeutic effects represent the most challenging fight. Doxorubicin (Dox) is a potent chemotherapeutic drug widely applied in human cancer treatments, including breast cancer [140]. Principal side effects occurring during Dox treatments are cardiac damage and bone marrow suppression, limiting the clinical application of Dox [141]. Encapsulation of Dox with a peptide-targeted liposomal nanosystem represents a successful approach to enhance its anti-tumor effect, reduce its cardiac toxicity, and changing its in vivo distribution [142,143].

In a recent work [144], the impact of modifying nanoparticle design parameters (linker length and peptide density) on the therapeutic efficacy of peptide-targeted liposomal doxorubicin nanoparticles against HER2+ breast cancer was also investigated, both in vitro and in vivo. The authors used the previously studied [135,137] cyclic HER2-binding peptide (HER2pep) as a targeted ligand and doxorubicin (Dox) as the chemotherapeutic agent since it is frequently used for treating HER2+ breast cancer [38]. In order to synthesize Dox-loaded HER2 receptor-targeted nanoparticles (TNPHER2pep), HER2pep was first conjugated to three lysine residues (K3) with an EG2 spacer. Then the obtained HER2pep-EG2-K3 was linked to palmitic acid lipid tails via an EGn linker (n = 8 or 18) to provide the HER2pep-K3-([EG18]/[EG8])-lipid. Simultaneously, a Dox–lipid conjugate was prepared by conjugating Dox to DPPE-GA (1,2-dipalmitoyl-sn-glycero-3-phosphoethanolamine-N-glutaryl) lipid by means of an acid-labile hydrazone bond in order to prevent premature Dox release and thereby non-specific toxicity, since Dox is preferentially released in an acidic environment (pH = 5.5). Finally, a Dox prodrug-loaded TNPHER2pep formulation was obtained incorporating peptide–lipid and drug–lipid conjugates into a liposomal platform with other lipid components at specific molar ratios. The authors conducted in vitro tests to assess the effectiveness of the HER2-targeted nanoparticles in selectively killing HER2-positive breast cancer cells. They evaluated the in vitro cytotoxicity of free Dox and Dox prodrug-loaded TNPHER2pep on SK-BR-3 (HER2+) and MCF-7 (HER2-) cell lines. They found that 1% TNPHER2pep[EG18]Dox obtained a maximum cytotoxic effect on HER2+ cells at ~5 μM Dox-equivalent concentration with an IC50 of ~2.5 μM, while it had no significant cytotoxicity against the HER2-negative MCF-7 cells even at 10 μM Dox equivalent concentration, indicating its high specificity for HER2-positive cells. The in vitro cellular uptake of TNPHER2pep formulations of various peptide densities (0.1–4%) was investigated by flow cytometry on HER2-positive human (BT-474 and SK-BR-3) and neu-positive mouse breast cancer cell lines; with MCF-7 as the negative control. The cellular uptake studies demonstrated that the EG8 linker and 1–2% peptide densities are the optimal parameters that result in enhanced cellular uptake both in human HER2+ and neu+ mouse breast cancer cells compared to the non-targeted nanoparticles. Tissue biodistribution and tumor cellular uptake were studied by using an in vivo MMTV-neu transplantation model and TNPHER2pep with different density and linker length also including non-targeted nanoparticles (NP). The in vivo studies showed that TNPHER2pep[EG8]Dox at 0.2–0.5% peptide density and TNPHER2pep[EG18]Dox at 0.35–0.5% peptide density significantly inhibited tumor growth compared to controls limiting systematic toxicity related to free Dox, while their accumulation levels in tumor tissue were comparable to that of non-targeted nanoparticles.

Several attempts have been made to improve the targeting selectivity of liposomal Dox carriers towards cancer cells. Sofou et al. [145], conducted a proof-of-principle study in vitro [146,147] in which they demonstrated the efficacy of pH tunable lipid vesicles, labeled with biotin as the targeting ligand, to mask and unmask functional groups based on the pH value of their surrounding environment, thus resulting in pH-dependent binding and internalization by streptavidin-presenting cancer cells. Starting from this study, the same authors functionalized their pH tunable lipid vesicles with an anti-HER2 peptide to enhance the targeted delivery of Dox to cancer cells [145]. This targeted peptide liposomal system successfully masked the HER2-targeting peptide in bloodstream circulation until reaching the tumor interstitium, where the targeting moiety was unmasked, allowing for specific binding to cancer cells. The vesicles then released Dox in a pH-mediated manner, resulting in high killing efficacy. 

The pH tunable vesicles were made up of 21PC, DSPS, and cholesterol (8.6:0.9:0.5 mol ratio), with 3.8 mol% of DSPE-PEG and DSPC/cholesterol-based vesicles (7/3 mol ratio) with 4.8 mol% DSPE-PEG. Dox was encapsulated into lipid vesicles by means of ammonium sulphate gradient method and the specific cancer cell targeting was mediated by a KCCYSL peptide. The peptide was synthesized using standard Fmoc solid-phase peptide synthesis protocols and included at a 0.2% mol on the vesicles.

In vitro investigations showed that the anti-HER2 pH tunable vesicles exhibited a 233% increase in binding to HER2-overexpressed BT474 breast cancer cells, with significant internalization. In vivo investigations on subcutaneous BT474 nude mice xenografts reported a 159% reduction in tumor volumes in targeted pH tunable systems compared to non-targeted vesicles. These targeted systems also demonstrated superior tumor control compared to targeted vesicles lacking the unmasked property.

The antiHER2/neu peptide (AHNP) is a small peptide derived from the trastuzumab monoclonal antibody, targeting the HER2/neu receptor. This peptide exhibits a strong affinity (300 nM) for binding to the HER2/neu receptor and effectively inhibits its kinase activity [148]. The conjugation of AHNP to drug delivery systems results in effective targeting and internalization into HER2/neu-positive cells. The anti-HER2/neu peptide, (FCDGFYACYADV) with three glycine amino acids serving as a spacer, was employed as a targeting ligand in the development of a new formulation of PEGylated liposomal doxorubicin (PLDs). This approach aimed to combine the tumor targeting properties of the AHNP peptide with the drug delivery characteristics of PLDs [149]. The authors investigated the effects of AHNP functionalization and density on several factors, including breast tumor cell uptake, selective cytotoxicity, inhibition of tumor growth, and finally DOX biodistribution within the tissues. Targeted liposomes were prepared using the post-insertion method [150], and AHNP was covalently linked to distearoyl-N-(3-carboxypropionoyl poly (ethylene glycol) succinyl) phosphatidylethanolamine (DSPE-PEG-COOH) micelles by the formation of an amide bond between the carboxyl group of lipids and the amine group of glycine residue of the spacers [151]. Doxorubicin was encapsulated within liposomes using the ammonium sulphate gradient technique [152], achieving an efficiency of 95%. The investigation of in vitro cytotoxicity revealed a notable reduction in IC50 values for both SK-BR-3 and TUBO HER2-overexpressing cells when using AHNP-peptide functionalized PLD containing 200 ligands, in comparison to formulations with 25, 50, or 100 ligands. Additionally, the in vitro studies carried out on these two cell lines demonstrated a higher uptake of liposomal formulations with the highest ligand density. 

The in vivo findings align with previous studies regarding the anticancer efficacy of AHNP when combined with various therapeutic agents and demonstrated the ability of AHNP to effectively delay tumor growth [148,153,154]. The utilization of peptide-targeted liposome systems in BALB/c mice bearing TUBO tumors effectively inhibited tumor growth rate dependent on the density of AHNP ligand, resulting in a reduction in tumor size. This effect was particularly enhanced in formulations containing 100 and 200 ligands. The conjugation of the peptide at the distal end of the PEG, the presence of free PEG-COOH, and the utilization of long-chain PEG improved the circulation and biodistribution of the targeted liposomes [155,156,157]. As a result, peptide-targeted liposomes showed superior specificity, safety, and circulatory behaviour compared to non-targeted liposomes. 

Self-assembled supramolecules represent a promising category of materials that are increasingly gaining attention for a wide range of biomedical applications, particularly in the field of drug delivery. These self-assembled structures typically consist of single monomers that are interconnected through weak, noncovalent interactions, including electrostatic, hydrophobic, and van der Waals forces, as well as hydrogen bonds and π-π stacking forces [158]. The large range of noncovalent interactions gives rise to distinct and well-defined structures with various properties and multiple functions. This versatility, coupled with the ease of synthesis and functionalization, as well as their biocompatibility, has led to the utilization of self-assembled peptide nanostructures and hydrogels in therapeutic and pharmaceutical applications [159]. 

Based on the “in vivo self-assembly” strategy for in situ construction of nanomaterials [160,161,162,163,164], Kit S. Lam et al. [165] developed a smart supramolecular peptide, BP-FFVLK-YCDGFYACYMDV. This peptide possesses the ability of self-assembling into nanoparticles (NPs) in aqueous environments and in blood circulation, and subsequently transforming into nanofibers (NFs) upon binding to HER2 receptors at tumor sites. In this study, the authors presented HER2-mediated, peptide-based, non-toxic transformable nanoparticles and demonstrated their efficacy as a monotherapy in HER2-positive breast cancer xenograft models. The peptide undergoes a structural transformation that effectively inhibits HER2 dimerization, thereby preventing signaling pathways and cell proliferation, leading to apoptosis of cancer cells. 

The transformable peptide monomer structure (BP-FFVLKYCDGFYACYMDV) consists of three functional domains: (a) the bispyrene BP moiety which exhibits hydrophobic properties inducing the formation of micellar NPs, (b) the KLVFF β-sheet forming peptide [166,167,168], and (c) the YCDGFYACYACYMDV disulphide cyclic peptide which serves as the HER2-binding domain [135,153,169]. The self-assembly property of the peptide monomer and its transformation from nanoparticles to nanofibers were confirmed through various techniques, including transmission electron microscopy (TEM), aggregation-induced emission (AIE), and dynamic light scattering (DLS). In vitro evaluation of the NPs’ cytotoxic effects was performed on SKBR-3 and BT474 breast cancer cell lines and on MCF-7/C6 cells. The NPs demonstrated a dose- and time-dependent inhibition of HER2 dimerization, promoting conversion of HER2 from dimers to monomers and blocking the phosphorylation of Erk, Mek, and Raf-1. In vivo therapeutic efficacy of the NPs was tested in MCF-7/C6 HER2+ breast cancer bearing mice. After administering the NPs eight times every other day via tail vein injection continuously for 40 days, the tumor was completely eliminated with no recurrence observed. The therapeutic effectiveness of NPs1 against HER2-positive tumors was also assessed in HER2-positive BT-474 and SKBR-3 xenograft models. Remarkably, the administration of NPs1 resulted in a highly favorable response, with nearly complete elimination of the BT474 tumor and complete eradication of the SKBR-3 tumor by day 40 with no side effects.

Among various nanoparticles developed and investigated in cancer treatment, iron oxide nanoparticles (IONPs) emerge as particularly interesting owing to their remarkably diverse range of applications. The combination of biocompatibility, biodegradability, optimal size, and unique properties such as superparamagnetism make them highly promising. They serve as contrast agents for MRI imaging in diagnostic or theragnostic applications, and can even be employed for therapeutic hyperthermia treatment [170,171].

One important aspect of optimizing this class of nanocarriers involves the incorporation of a specific target ligand, aiming to achieve receptor-mediated internalization within tumors.

Zhang et colleagues [172] reported the application of small (∼30 nm) and stable iron oxide nanoparticles (IONPs) for the purpose of the targeted delivery of paclitaxel (PTX) to HER2/neu-positive breast cancer cells employing an anti-HER2/neu peptide (AHNP) as the targeting ligand. This study demonstrated the excellent stability of these nanoparticles in biological environments, their successful targeting of tumors in live mice, as well as the selective elimination of HER2/neu-positive breast cancer cells. AHNP, derived from the trastuzumab monoclonal antibody, was covalently linked to polyethylene glycol (PEG) monolayer-coated IONPS (IONP-PEG-NH2) to provide effective targeting and internalization of the nanoparticles into HER2/neu+ cells. Additionally, carboxymethylated-β-cyclodextrin (CM-β-CD) was conjugated onto IONPS to achieve PTX loading via a hydrophobic interaction between PTX and the cyclodextrin moiety [173,174]. The drug loading percentage was approximately 14.7%, and HPL-MS analysis quantified around 170 AHNP molecules per NPs. The IONP-PTX-AHNP formulation exhibited a uniform core size of approximately 12 nm, as observed through TEM investigations. Furthermore, dynamic light scattering (DLS) analyses conducted in PBS indicated a hydrodynamic diameter of 30.2 nm, while the slightly negative zeta potential of the particles was found to align well with the intended targeting purpose [175,176]. To evaluate the in vivo targeting capabilities of AHNP-conjugated IONPs, experiments were conducted using a human HER2/neu+ SK-BR-3 breast cancer xenograft mouse model. To facilitate imaging, a near-infrared dye (Cy5.5-NHS) was conjugated to the free amine groups of IONPs after AHNP conjugation. In vivo tests demonstrated that AHNP-conjugated IONPs accumulated in the tumors within 6 h after the injection and remained retained in the tumor for at least 72 hours. Detection was also observed in other organs such as the brain, liver, spleen, and spine. However, the fluorescence signals began to diminish after 96 h, suggesting the gradual elimination of the conjugated IONPs from these organs. In vitro investigations using two distinct human breast cancer cell lines with high (SK-BR-3), and low (MDA-MB-231) HER2/neu expression levels revealed that uptake was specific to HER2/neu+ cells. The SK-BR-3 cells exhibited a four-fold increase in NP uptake compared to the MDA-MB-231 cells. Moreover, this differential uptake was accompanied by enhanced cell killing, specifically observed in SK-BR-3 cells.

Nanoparticles for therapeutic and diagnostic purposes should be able to deeply penetrate inside the desired tissue, ensuring uniform nanoparticle distribution. In this regard, targeted small nanoparticles characterized by elongated shapes, demonstrate superior performance due to their hydrodynamic features [177,178,179]. 

Holler et al. [180] proposed a new mini nanodrug consisting of PMLA (poly(β-L-malic acid), which is conjugated with several HER2-specific antisense oligonucleotides (AONs) [181], along with several 12-mer mimetic peptides that specifically target the HER2 receptor. Additionally, a series of tri-leucine residues are incorporated to facilitate cytoplasmic escape following endosomal uptake by recipient cells. The hydrodynamic radius of the nanoconjugate corresponds to the extension of the polymer backbone, while the peptide and oligonucleotide substituents along the chain are oriented orthogonally to the polymer structure and do not contribute significantly to the measured radius.

The anti-HER2 mimetic peptide, AHNP, NH_2_-YCDGFYACYMDV-NH_2_ contains an exocyclic disulfide bridge, spontaneously formed during sulfhydryl oxidation in air [136]. A pre-conjugate P/LLL(40%)MEA(10%) was prepared by amidation of NHS-activated polymalic acid (P) using (L-leucine)3 (LLL) and 2-mercaptp-1-ethylamine (MEA) [182]. Two of eight carboxylic groups of starPEG were amidated with NH_2_-PEG200-AHNP. In the same reaction mixture, Mal-PEG3400-NH was added, obtaining StarPEG(PEG200-AHNP2)-PEG-Mal. Successively, purified Mal-PEG-3400-starPEG was thio-alkylated by reaction with –SH groups of pre-conjugate. Once reaction was completed, the remaining sulfhydryl groups of P/LLL(40%)/PEG3400-starPEG(PEG200-AHNP)2 (2%) were conjugated with 3-(2-pyridyldithio)-propionyl-amino-AON (PDP-AON) to yield P/LLL(40%)/PEG3400-starPEG(PEG200-AHNP)2(2%)/AON(1.5%). To validate the target activity of AHNP, imaging tests were conducted using Alexa Fluor 680 and the Xenogen IVIS Imaging System. These tests involved analysing the uptake and accumulation of the nanoconjugate in target tissues of nude mice with BT474 human breast flask tumors. The imaging experiments confirmed both tumor recognition and uptake, indicating binding to the HER2 receptor. The observed high fluorescence intensities in the liver and kidney, along with the comparatively low intensity in the spleen, align with the expected function of clearance pathways, while the absence of fluorescence in the lung, heart, brain, and skin suggested the absence of toxic side effects of the nanoconjugate. In vivo investigations demonstrated the growth inhibition of HER2-positive breast cancer. After treatment with the nanodrug, the tumor size was significantly reduced and became undetectable.

The p160 peptide emerges as a highly promising candidate with notable targeting properties. Discovered through random peptide phage display, the p160 peptide (VPWMEPAYQRFL) is a linear dodecapeptide that exhibits specificity for breast cancer cell lines. Notably, it demonstrates the capability of internalizing in various breast cancer cells, including MDA-MB-435 and MCF-7. Importantly, it does not exhibit internalization in healthy human umbilical vein endothelial cells (HUVECs) [183,184,185]. 

Lavasanifar and co-workers developed an analogue of p160 peptide, namely p18-4 (WXEAAYQRFL), with enhanced selectivity for cancer cells compared to normal cells and improved stability in biological fluids [186]. Subsequently, in a separate study, they conducted an evaluation of the engineered p18-4 peptide as a ligand for targeting breast tumors using liposomal nanocarriers loaded with DOX [187]. To achieve this, three different coupling methods were employed: conventional coupling, post conjugation, and post insertion. The impact of the peptide coupling method on the cytotoxicity and cellular uptake of liposomal DOX formulations was then assessed. In the conventional method, the synthesized p18-4-PEG-DSPE lipid conjugate was mixed with PEG-DSPE to form liposomes, and subsequently, DOX was loaded into the liposomes. The conventional method may lead to the generation of liposomes with the peptide also embedded within the vesicular structure. On the other hand, the post conjugation method involves a reaction between carboxylic groups on the liposomal bilayer and the N-terminal of the peptide. This method ensures that all the conjugated peptide is displayed on the liposomal bilayer. The post-insertion method is based on the exchange of PEG-DSPE peptide conjugate from micelles with the liposomal phospholipid membrane. This exchange occurs at temperatures higher than the phase transition temperature of the phospholipid [188,189,190]. The conventional and post conjugation techniques did not have a significant impact on the encapsulation and release rate of DOX when compared to untargeted liposomes. However, liposomes prepared using the post-insertion technique exhibited lower levels of DOX encapsulation and release from the carrier. It is possible that the post-insertion process perturbed the liposomal membrane, resulting in the loss of loaded DOX. Peptide conjugation efficiency and density were 35% and 0.3% mole, respectively, for all the adopted methods. P18-4 liposomes prepared using conventional or post conjugation methods exhibited an encapsulation efficiency of over 95% for DOX, while the PI method yielded liposomes with a considerably lower DOX encapsulation efficiency, measuring less than 30%. In vitro experiments revealed that the uptake of p18-4 modified liposomes, obtained by the conventional or post conjugation method, by MDA-MB-435 cells was comparable to each other, but significantly higher than that of unmodified liposomal DOX formulations (1.8- and 1.6-fold increases, respectively). This increase can be attributed to the specific interaction between the p18-4 peptide and MDA-MB-435 cell receptor. No enhancement in the uptake of liposomal DOX by MDA-MB-435 cells was observed with liposomes produced using the post-insertion method.

In a subsequent study, the same authors investigated the selectivity in vitro and anticancer activity in vivo of the optimized p18-4-modified liposomal DOX [191]. The selection of the appropriate peptide density played a key role in designing effective, actively targeted liposomal drug carriers. Hence, in order to minimize non-specific interactions of drug carriers and improve the targeted delivery of encapsulated cargo to specific cells, the density of the targeting ligand on the drug carrier was optimized. Targeted liposomes with varying densities of p18-4 peptide were prepared using the conventional method previously described. Two distinct formulations of p18-4-modified DOX liposomes referred to as p18-4 liposomes HD and p18-4 liposomes LD with a molar peptide density of 1.5 and 0.3 mol%, respectively, were developed. The degree of peptide conjugation did not impact the efficiency of DOX conjugation into liposomes which was greater than 95%. The in vitro tests demonstrated that p18-4 peptide-modified HD liposomal DOX exhibited a 2.4-fold decrease in IC50 in MDA-MB-435 cells and a 5-fold decrease in IC50 in MCF-7 cells compared to unmodified liposomes. Conversely, LD liposomes exhibited a lesser decrease (1.6-fold and 2.2-fold) in IC50 values in the same breast cancer cell lines. In an evaluation of the in vivo anticancer activity in NOD-SCID mice bearing MDA-MB-435 xenografts, LD liposomal DOX demonstrated superior anticancer activity compared to HD and unmodified liposomal formulations. Mice treated with LD liposomal DOX exhibited a remarkable 4.8-fold reduction in the mean relative tumor volume when compared to non-targeted DOX liposomes. This result could be attributed to an unfavorable biodistribution profile of the HD formulation and its potential uptake by phagocytic cells, which may result in the generation of an inflammatory response within the tumor.

Polymeric micelles, which form a core–shell structure using amphiphilic copolymers improve solubility and bioavailability of poorly soluble drugs [192] preventing undesired degradation. Polymeric micelles loaded with drugs through hydrophobic interactions offer several advantages, including ease of preparation and applicability for several hydrophobic drugs. However, their use is also associated with certain drawbacks, such as inconsistent drug release rates and low cellular uptake [193,194]. However, drug loading via covalent binding, which enables controlled release, and the application of targeted ligands to enhance cellular uptake, have the potential to overcome these challenges. Taking these factors into consideration, Jae-Woon Nah and his research team [195] prepared novel polymeric micelles to develop a targeted and redox-sensitive drug delivery system (HPTOC-DOX) for two chemotherapeutic drugs, Tocopherol (TP), and Doxorubicin (DOX). The synthesized polymeric micelles consist of a backbone of O-carboxymethyl chitosan (OCMCh), a chitosan derivative modified with CH_2_COOH at the C6 position. This modification enhances its solubility in aqueous solutions while retaining beneficial properties such as bioactivity, antibacterial activity, and stability [196,197,198]. The synthesis of polymeric micelles (HPTOC-DOX) involved three sequential steps: first, α-tocopherol (TP) was conjugated to O-carboxymethyl chitosan (TOC); next, DOX was linked to TOC through a redox-sensitive dithiobis-succinimidyl propionate (DTSP) [199]; and finally, a targeting ligand (anti-HER2/neu peptide-polyethylene glycol [PEG]; HP) was conjugated to TP and DOX-conjugated polymeric micelles (TOCDOX) to promote site-specific targeting. The synthesized HPTOC-DOX incorporating the anti-HER2/neu targeting peptide (epitope form, LTVSPWY) exhibited the following effects: a synergic effect resulting from the conjugation of TP and DOX, and a site-specific delivery of the anti-cancer drugs. The HPTOC-DOX micelles showed a superior therapeutic effect compared to TOC-DOX. This was observed both in vitro, using SK-BR-3 cells, and in vivo, in mice bearing SK-BR-3 tumors. The results provided clear evidence that cellular uptake of HPTOC-DOX occurred through receptor-mediated endocytosis. The interaction between the anti-HER2-targeting peptide and the overexpressed HER2/neu receptor on the surface of SK-BR-3 cells significantly improved the cellular uptake of the micelles, thereby enhancing their therapeutic efficacy. Furthermore, the synergistic effect of TP and DOX was demonstrated by the substantial low cell viability observed after treatment with a very low concentration (0.078125 μg/mL) of both TOC-DOX and HPTOC-DOX. Additionally, a hemolysis assay confirmed the safety of the drug carrier for systemic administration.

Hyaluronic acid (HA) is a biodegradable anionic natural polysaccharide mostly found in connective tissues [200]. It serves as a ligand for the CD44 receptor, which is often overexpressed in various tumor cells [201]. The use of hyaluronic acid-g-poly(L-histidine) (HA-PHis) copolymer-based micelles represents an effective strategy for enhancing cellular uptake and expediting the efficient transport of doxorubicin (DOX) to the cytosol. This, in turn, yields notable enhancements in therapeutic efficacy [202]. In their research, Cheng and colleagues developed pH-sensitive mixed copolymer micelles using a combination of HA-g-poly(L-histidine) (PHis) and d-α-tocopheryl polyethylene glycol 2000 (TPGS2k) amphiphilic polymer with the aim of delivering DOX to drug-resistant breast cancer cells [203]. To further enhance their targeting capabilities, the micelles composed of HA-PHis and TPGS2k copolymers were modified by incorporating a HER2 peptide on their surface [204]. This peptide acts as a specific ligand for breast cancer cells, enabling precise dual-targeting efficacy and extensive penetration into breast tumors. HER-2 peptide-modified micelles were synthesized by conjugating the HER2 peptide (YCDGFYACYMDV) to TPGS2k, forming pep-TGS2k. This conjugation was achieved through a reaction involving the carboxyl group of PEG and the amino group of the HER2 peptide. Dox loading onto LHA-Phis/pep-TPGS2k mixed micelles (PHTM) was achieved using the ultrasonication method. Compared with DOX-loaded HA-Phis/TPGS2k mixed micelles (HTM), the copolymer micelles incorporating the HER2-specific peptide showed higher cytotoxicity and more efficient uptake in MDA-MB-231 cells overexpressing CD44 and HER2 receptors. Following a 4-hour incubation period, PHTM exhibited a 1.3-fold increase in the uptake efficiency of DOX compared to HTM. The cellular uptake and distribution of DOX from the developed micelles in MDA-MB-231 cells were further examined using confocal laser scanning microscopy (CLSM). After the 4-hour incubation, the fluorescence intensity of PHTM was found to be stronger than that of HTM. To further elucidate cellular uptake and distribution, the DOX-loaded micelles were examined in MDA-MB-231 cells using confocal laser scanning microscopy (CLSM). After the 4-hour incubation, the fluorescence intensity emitted by PHTM exhibited greater intensity when compared to HTM. These findings align with the observation that the improved cellular uptake of PHTM can be attributed to both the interaction between HA and CD44 receptors, as well as the interaction between the peptide and HER2 receptors. The real-time biodistribution of PHTM in MDA-MB-231 tumor-bearing mice was assessed using a noninvasive near-infrared (NIR) optical imaging technique. The evaluation demonstrated a significantly stronger fluorescence intensity (1.76-fold increase) in the tumor tissue after 12 hours of administration, compared to HTM. These results provide clear evidence that the HER2 peptide-modified HA-based micelles have the ability to enhance the accumulation of anticancer drugs within tumors. Interestingly, no fluorescence signals were detected in the heart, indicating that the developed micelles have the potential to effectively reduce the cardiotoxicity induced by DOX. For the assessment of PHTM’s in vivo antitumor effectiveness, the tumor volume in mice harboring tumors was tracked. The findings revealed a noteworthy decrease in the average tumor weight within the PHTM-treated cohort, registering a 2.31-fold reduction compared to free DOX and a 1.69-fold reduction compared to HTM. These results highlight the considerable enhancement in antitumor efficacy achieved through the utilization of HA-based micelles decorated with the HER2 peptide. Table 3 illustrates HER2-targeted peptide-based nanodevices for breast cancer treatment.

### 4.2. Application of HER2-Targeted Peptides for Tumor Imaging

Regulatory peptides form a class of molecules that have been designed for precise cancer targeting. They are currently attracting growing attention due to their favorable comparison to antibody-based immunotargeting approaches. Peptides offer numerous advantages including their compact size, which facilitates efficient diffusion through tissues and enables easy access to targets. Furthermore, peptides can be readily synthesized, and their radiolabeling process is straightforward and adaptable. Numerous studies have demonstrated significant overexpression of several peptide receptors in various types of cancers, in contrast to their relatively low expression in normal physiological organs. The pronounced overexpression of regulatory peptide receptors has led to increased interest in exploiting these peptides for cancer imaging purposes [205,206]. 

HER2 has emerged as a crucial molecular marker in breast cancer, as it is amplified in approximately 30% of cases, and it holds significant importance for both prognosis and treatment strategies in breast cancer. Molecular imaging of HER2-positive breast cancer offers a valuable tool in the management of the disease, providing essential insights for personalized treatment decisions.

The LTVSPWY peptide, extensively investigated and recognized as a HER2 target ligand, exhibits the ability to facilitate the internalization of particles into breast cancer cells via peptide-mediated endocytosis [193,207,208,209].

Li-Yong Jie et al. [209] synthesized LTVSPWY PEGylated chitosan-modified magnetic nanoparticles using Fe_3_O_4_-based MRI (magnetic resonance imaging) contrast agents. These nanoparticles were developed as carriers for early detection of HER2-overexpressing tumors and further diagnostic applications. The preparation of magnetic nanoparticles involved the utilization of the solvent diffusion method, wherein oleic acid served as a dispersant to achieve a stable aqueous dispersion of Fe_3_O_4_ nanoparticles. The adsorption of oleic acid onto the surface of Fe_3_O_4_ nanoparticles facilitated the coating process using solid lipid material monostearin. Furthermore, in order to optimize the potential of lipid nanoparticles as an imaging agent, their surface was modified by incorporating a ligand. This modification enables a more effective and precise targeting of cancer cells, ensuring enhanced selectivity and affinity. The formation of LTVSPWY-PEG-CS-modified magnetic nanoparticles was achieved by exploiting the charge interaction between the positively charged LTVSPWY-PEG-CS and the negatively charged magnetic nanoparticles. The efficacy of these modified magnetic nanoparticles as MRI contrast agents for tumor targeting was investigated through in vitro studies using a coculture system comprising SKOV-3 cells overexpressing HER2 marker and HER2-negative A549 cells. Interestingly, the data confirmed a strong and specific binding of peptide-lipid-modified nanoparticles to SKOV-3 cells, due to endocytosis mediated by the LTVSWPY peptide. To assess the in vitro cytotoxicity of peptide-modified magnetic nanoparticles, the MTT test was conducted on SKOV-3 and A549 cell lines. Remarkably, the results revealed that the cell viability of both cell lines remained above 80%, implying that these nanoparticles exhibit minimal toxicity as carriers for cancer diagnostics. The biodistribution analysis of modified magnetic nanoparticles, carrying a near-infrared-emitting DiR (1,1′-dioctadecyl-3,3,3′,3′-tetramethyl indotricarbocyanine iodide) probe, was conducted in a nude mouse model of a SKOV-3 xenograft. Notably, LTVSPWY-PEG-CS-modified nanoparticles exhibited a higher accumulation in the tumor region within just one hour, as compared to PEG-CS-modified and chitosan-modified magnetic nanoparticles. The PEG modification enhanced blood circulation time in vivo, preventing undesired protein binding. As a result, treatment with LTVSPWY-PEG-CS-modified DiR-loaded magnetic nanoparticles enabled faster and more efficient in vivo identification and diagnosis of tumors.

Quantum dots (QDs) are remarkable semiconductors or fluorescent nanocrystals that possess unique physical and chemical properties. QDs offer numerous advantages over other materials, including enhanced photo and chemical stability, higher quantum yield, and minimal toxicity. Furthermore, QDs can be modified by attaching specific drugs or bioactive molecules to their surface, making them attractive targeting ligands for improved cell targeting and theragnostic applications. These platforms hold great potential for diverse biological and biomedical applications, including biomedical imaging, and drug delivery [210,211,212,213]. Recent studies have demonstrated the potential of quantum dot (QD)-based probes for preclinical applications, garnering significant attention in the scientific community [214,215,216,217,218]. Surface modification of quantum dots (QDs) with biomolecules, especially HER2-targeting peptides, is essential for using QDs as fluorescent probes for HER2 receptor imaging. This strategic modification allows for the specific targeting of HER2 receptors for diagnostic and research purposes.

Schneider and co-workers designed CuInZn_x_S_2+x_ quantum dots (ZCIS QDs) as fluorescent nanoprobes, which were functionalized with a HER2-targeting peptide to selectively target HER2-positive cancer cells [219]. They developed a novel coating technique using PMAO (poly(maleic anhydride-alt-1-octadecene)) polymer to encapsulate ZCIS QDs, which resulted in nanocrystals with remarkable colloidal stability, a high photoluminescence quantum yield (35%), and minimal cytotoxicity. ZCIS QDs were prepared using a modified version of the synthetic procedure they had previously developed [220]. The synthesis involved the utilization of the Zn(OAc)_2_–OA complex to introduce a ZnS shell around the CuInS_2_ QDs core. To achieve specific targeting of HER2 receptors, they conjugated the LTVSPWY HER2-binding peptide to the surface of the QDs after surface modification with 2,2-(ethylenedioxy)bis(ethylamine) linker (diaminoPEG).

The resulting ZCIS@PMO-diaminoPEG-peptide presented a hydrodynamic diameter of 39 nm, with a PDI (polydispersity index) value of 0.27, showing a narrow size distribution. Furthermore, stability studies of the nanocrystals demonstrated good colloidal stability with no visible signs of aggregation for at least 4 months in PBS buffer. The cell binding efficiency of peptide-functionalized QDs was assessed by means of flow cytometry following incubation of ZCIS@diaminoPEG-peptide QDs with both HER2-negative MSU1.1 and HER2-positive SKBR3 cells. These findings revealed that the peptide-linked QDs exhibited significantly higher binding and/or internalization by SKBR3 cells, as evident from their higher average fluorescence intensity compared to the control ZCIS@PMO QDs. Additionally, in vitro imaging experiments through confocal microscopy on SKBR3 cells confirmed the cellular internalization of peptide-linked quantum dots (QDs) by HER2-positive cancer cells. After only 30 min of incubation, internalization of peptide-loaded QDs by SKBR3 cells was observed, indicating that QDs entered the cells through an endocytosis mechanism.

Wang et al. [221] have successfully obtained and validated two novel peptide probes targeting HER2, YLFFVFER (H6) and KLRLEWNR (H10), by using an efficient peptide screening strategy based on in situ single bead sequencing on a microarray. Two different peptide-modified QD nanosystems (H6-QDs and H10-QD probes) were developed for the purpose of breast cancer imaging and diagnosis. The affinity and specificity toward the HER2 receptor of the novel octapeptides H6 and H10 were confirmed by using HER2 high-expressing SKBR-3 cell lines as the model cells. These results demonstrated that both the peptides H6 and H10 bind to HER2-positive cells in a specific manner, indicating that HER2 protein is the specific target of these peptide ligands. The biodistribution of peptide-loaded QD nanoprobes was investigated in SKBR-3 xenograft mice using the small animal imaging system. The results revealed that H10 nanoprobes exhibited superior specificity and a shorter plasma half-life compared to H6 nanoprobes. Furthermore, the MTT assay showed lower cytotoxicity for both nanoprobes compared to QDs alone. These findings highlight the potential of the combined peptide-nanomaterials as highly promising probes for in vivo imaging applications. Table 4 summarizes HER2-targeting peptide-based nanosystems for diagnosis.

## 5. Conclusions

Breast cancer remains a significant global health concern, and HER2-positive breast cancer represents a particularly aggressive and clinically challenging subtype. HER2 protein plays a critical role in regulating cell growth and proliferation, and its overexpression is associated with increased cell proliferation, invasiveness, and poor prognosis in breast cancer. Over the years, there have been significant advancements in the treatment of this disease, and one of the most promising approaches is the use of molecular targeted therapy. 

This review highlights the significance of HER2 as a pivotal therapeutic target within breast cancer treatment. The comprehensive array of studies discussed in this review encompass the principal targeting approaches employing antibodies and peptides explored over the past decade, displaying promising outcomes and potential advancements in diagnosis and therapy of HER2-positive breast cancer.

The peptide-based targeting of HER2 has gained significant attention in recent years due to the low immunogenicity, low cost, long-term storage, and easy handling of peptides. 

Moreover, in addition to acting as targeting agents, peptides have also been employed for delivering drugs or imaging agents specifically to HER2-expressing cancer cells, opening new avenues for early detection, disease monitoring, and treatment.

However, despite the promising results of peptide-based targeting of HER2 in preclinical studies, challenges related to low target affinity, limited specificity, and metabolic instability due to enzymatic degradation need to be addressed. Further research is necessary to optimize the use of peptides as targeting agents for HER2-positive breast cancer, possibly through strategies involving suitable modifications of the peptide sequence.

Another promising opportunity lies in the application of nanotechnology for the development of HER2-targeted nanosystems, which could address current clinical challenges and advance breast cancer imaging, targeting, and therapy. 

The peptide-modified nanoparticle systems possess a distinctive set of properties including proper size, prolonged serum half-life, and ability to target specific cells that generate a powerful combination. These properties combined with the peculiarities of the tumor microenvironment foster an increased drug accumulation at the tumor site.

The biophysical and chemical properties of peptide ligands are critical factors in determining the therapeutic potential of peptide-functionalized nanoparticles for targeted cancer treatment and tracking. Improving the performance of peptide-targeted drug delivery systems through modifications in the peptide ligand’s structure, hydrophobicity, and density on the nanoparticles is crucial. These modifications can significantly improve the efficiency and specificity of drug delivery to target cells or tissues, maximizing therapeutic outcomes while minimizing off-target effects. Furthermore, advancements in nanoparticle engineering that allow for precise control over the density of peptides on the surface can further fine-tune the overall drug delivery process.

In conclusion, the employment of peptides as targeting agents for HER2-positive breast cancer presents a highly encouraging avenue for creating innovative and effective targeted treatments. Ongoing research in this field has the potential to significantly improve patient outcomes and advance the field of cancer therapy. By addressing the challenges and exploiting the potential of nanotechnology, peptide-based targeting approaches can further revolutionize breast cancer treatment, bringing more personalized and efficient therapeutic strategies.

## Figures and Tables

**Figure 1 nanomaterials-13-02476-f001:**
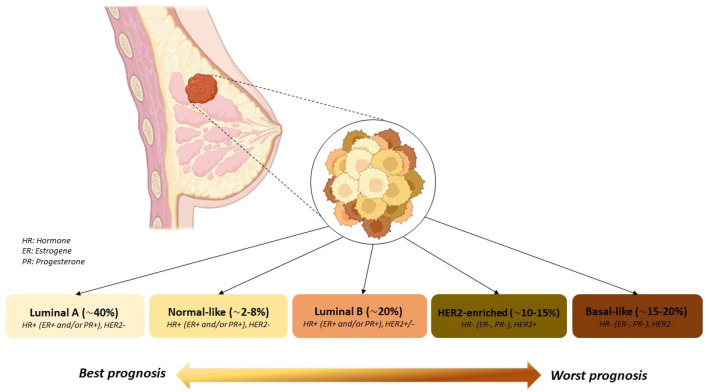
Breast cancer subtypes can be classified based on their hormone receptor expression: estrogen receptor (ER) progesterone receptor (PR), and human epidermal growth factor (HER2).

**Figure 2 nanomaterials-13-02476-f002:**
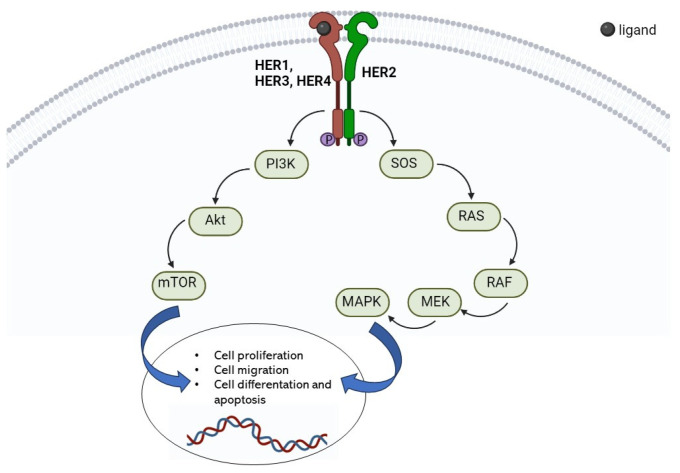
HER-2 signaling pathways. Ligand binding to the extracellular domain of the human epidermal growth factor receptor and subsequent dimerization leads to the phosphorylation of tyrosine residues within the intracellular domain of HER2. Phosphorylation of the tyrosine kinase domain prompts the activation of oncogenic signaling pathways such as the PI3K/AKT pathway and the RAS/MAPK pathway. These pathways play a critical role in regulating the transcription of genes responsible for driving cellular processes such as proliferation, migration, differentiation, and apoptosis.

**Figure 3 nanomaterials-13-02476-f003:**
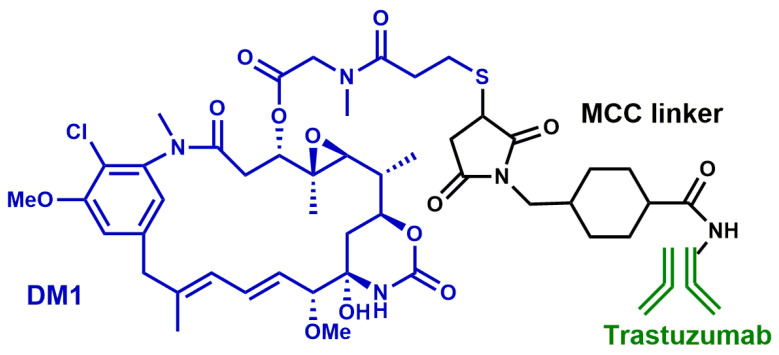
T-DM1 structure. T-DM1 is composed of the monoclonal antibody Trastuzumab (shown in green), which is conjugated to the potent tubulin polymerization inhibitor DM1 (N2’-deacetyl-N2’-(3-mercapto-1-oxopropyl)-maytansine (shown in blue) through a non-cleavable MCC (4-[N-maleimidomethyl]-cyclohexane-1-carbonyl) thioether linker (shown in black).

**Figure 4 nanomaterials-13-02476-f004:**
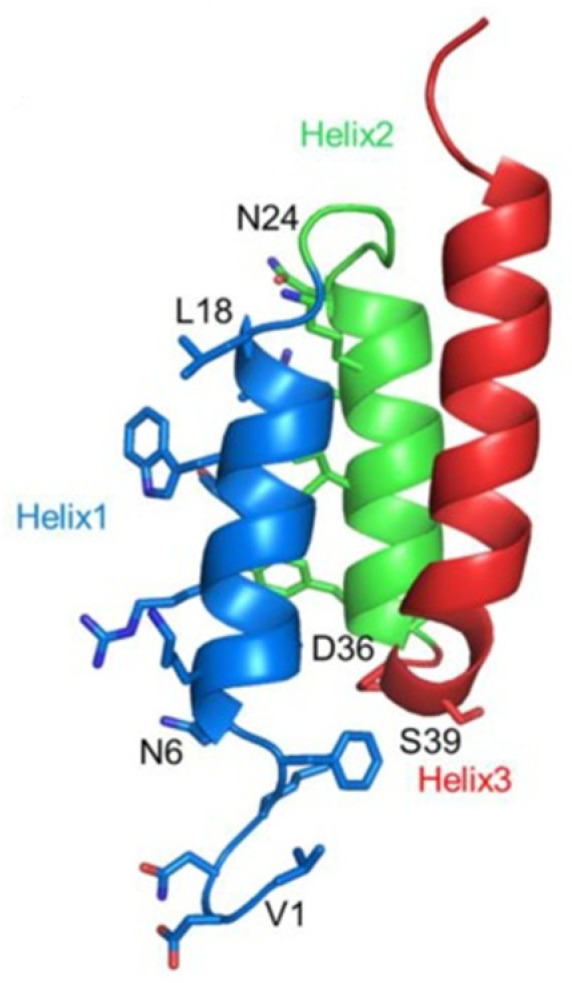
Structure of affibody Z(HER2:342). The affibody Z(HER2:342), comprising 58 amino acids, is structured into three distinct α-helices: helix1 (Asn6-Leu18), helix2 (Asn24-Asp36), and helix3 (Ala42-Gln55).

**Figure 5 nanomaterials-13-02476-f005:**
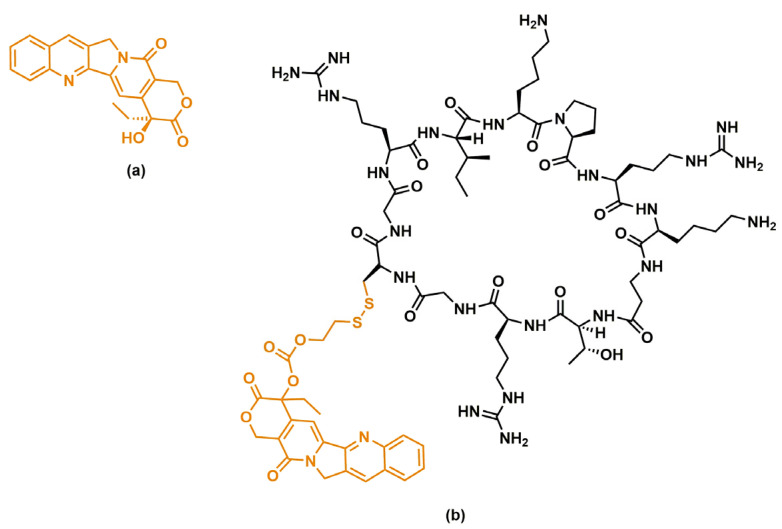
Structure of camptothecin (**a**) and structure of Conjugate I (Cyclo-GCGPep1-Camptothecin) (**b**).

**Figure 6 nanomaterials-13-02476-f006:**
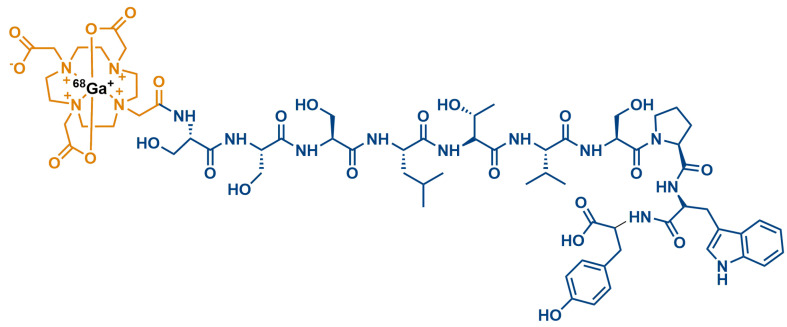
Chemical structure of the 68Ga-DOTA-(Ser)3-LTVSPWY peptide. The peptide (LTVSPWY) linked to a spacer consisting of three serine residues ((Ser)3) (blue), is conjugated to the radionuclide (68Ga) contained within the DOTA chelator (orange).

**Figure 7 nanomaterials-13-02476-f007:**
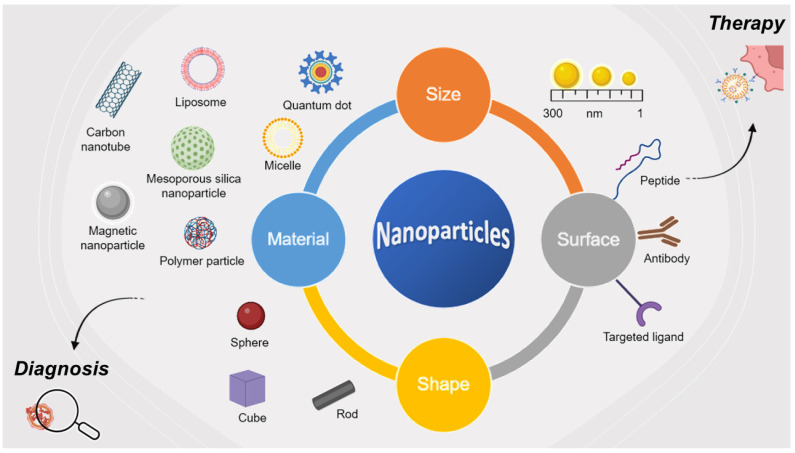
Schematic illustration of nanoparticle-based strategies for breast cancer diagnosis and therapy.

**Figure 8 nanomaterials-13-02476-f008:**
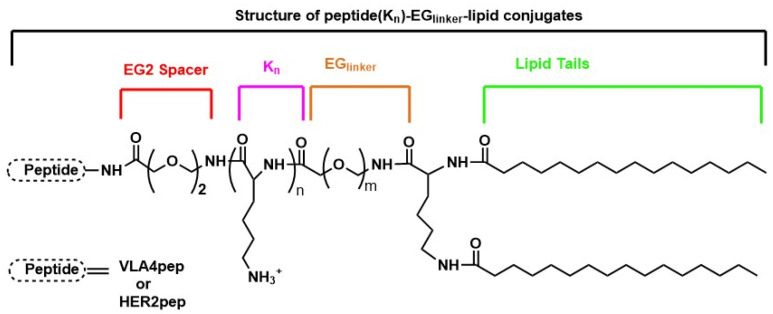
Structure of peptide (K_N_)–EG_linker_–lipid conjugates. The peptide–lipid conjugates consist of a targeting peptide, an EG2 spacer (red box), a short oligolysine chain (KN, where K refers to lysine and N is the number of repeat units) (pink box), an EG peptide linker (orange box), and two hydrophobic fatty acid chains (green box).

**Figure 9 nanomaterials-13-02476-f009:**
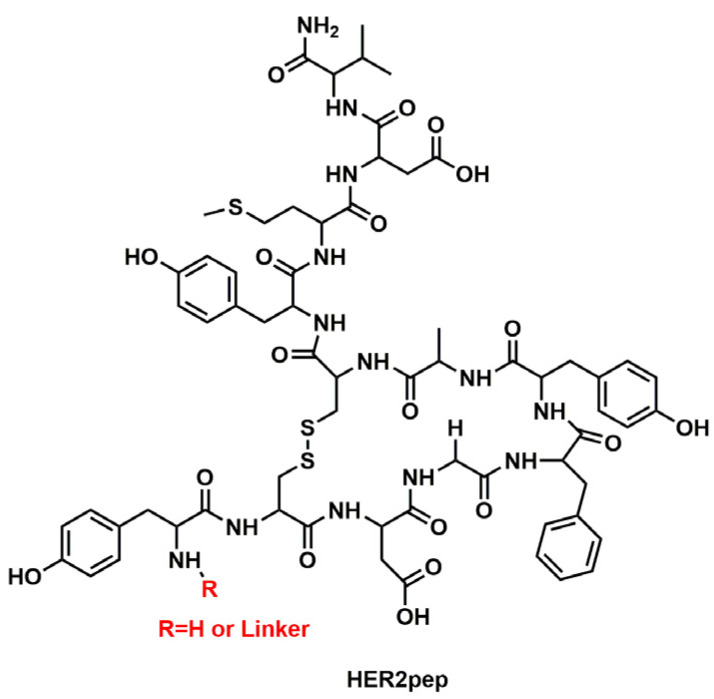
HER2pep structure. HER2pep is a cyclic peptide sequence comprising 12 amino acids, YCDGFYACYMDV, designed to selectively target the HER2 receptor.

**Figure 10 nanomaterials-13-02476-f010:**
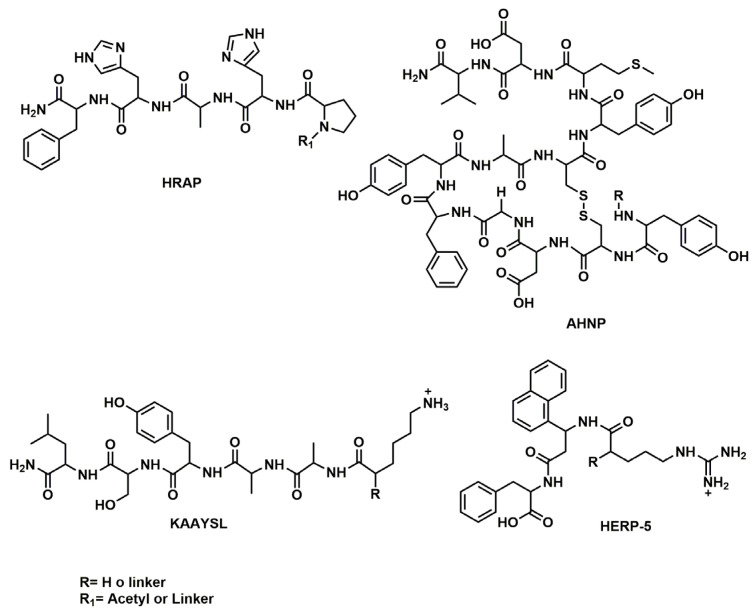
The chemical structures of HER2 antagonist peptides, HRAP, AHNP, KAAYSL, and HERP5.

**Table 1 nanomaterials-13-02476-t001:** Sequences of five potential HER2-targeting peptides.

No.	Sequence
Pep 1	RIKPRKGYTR
Pep 2	RIKRTNRYTR
Pep 3	RIRPTRRYTR
Pep 4	RIRPRNRYTR
Pep 5	RIRPRKGYTR

**Table 2 nanomaterials-13-02476-t002:** Sequences of HER2-binding peptides.

Sequence	Name
CF-KCCYSL-NH_2_	P(CC)
CF-KCGCYSL-NH_2_	P(CGC)
CF-KCGGCYSL-NH_2_	P(CGCG)
CF-KC_(Acm)_C_(Acm)_YSL-NH_2_	P(C_(Acm)_C_(Acm)_)
CF-KCSYSL-NH_2_	P(CS)
CF-KSCYSL-NH_2_	P(SC)
CF-KSSYSL-NH_2_	P(SS)
CF-KAAYSL-NH_2_	P(AA)
CF-GYYNPT-NH_2_	P(YY)
CF-KAAYSLGYYNPT-NH_2_	cP(AA)_P(YY)
CF-KSCYSLGYYNPT-NH_2_	cP(SC)_P(YY)
CF-YSLGYYNPT-NH_2_	P(short)_PYY
CF-TAKLYPGYANYS-NH_2_	scr_P(AA_YY)
CF-GYYNPTKAAYSL-NH_2_	cP(YY)_P(AA)
H-KAAYSLGYYNPT-NH_2_	Unlabeled cP(AA)_P(YY)
H-KSCYSLGYYNPT-NH_2_	Unlabeled cP(SC)_P(YY)

**Table 3 nanomaterials-13-02476-t003:** HER2-targeting peptides in drug delivery.

Peptide	Delivery System	Chemotherapeutic Agent	Clinical Studies	Ref.
WNLPWYYSVSPTC	Liposome	Capecitabine	In vitro studies	[134]
Cyclic YCDGFYACYMDV	Liposome		In vitro studies	[135]
HERP5	Liposome		In vitro studies	[136,137,138,139]
HRAP	Liposome		In vitro studies	[136,137,138,139]
KAAYSL	Liposome		In vitro studies	[136,137,138,139]
AHNP	Liposome		In vitro studies	[136,137,138,139]
YCDGFYACYMDV	Liposomal platform	Doxorubicin	In vitro and in vivo studies	[144]
GSG-(KCCYSL)	Liposomal vesicle	Doxorubicin	In vitro and in vivo studies	[145]
FCDGFYACYADV	Liposome	Doxorubicin	In vitro and in vivo studies	[148,149,153,154]
BP-FFVLK-YCDGFYACYMDV	Self-assembled nanoparticle	BP-FFVLK-YCDGFYACYMDV	In vitro and in vivo studies	[165]
AHNP	Iron Oxide Nanoparticle	Paclitaxel	In vitro and in vivo studies	[172]
YCDGFYACYMDV	Mini nanodrug	Morpholino antisense oligonucleotides (AONs)	In vitro and in vivo studies	[180]
WXEAAYQRFL	Liposome	Doxorubicin	In vitro and in vivo studies	[185,186,191]
anti-HER2/neu targeting peptide (epitope form, LTVSPWY)	Polymeric micelle	Doxorubicin	In vitro and in vivo studies	[195]
YCDGFYACYMDV	Copolymer-based micelle	Doxorubicin	In vitro and in vivo studies	[203]

**Table 4 nanomaterials-13-02476-t004:** HER2-targeting peptide-based nanosystems for imaging.

Peptide	Delivery System	Clinical Studies	Ref.
LTVSPWY	Lipid-modified Fe_3_O_4_-based magnetic nanoparticles	In vitro and in vivo studies	[209]
LTVSPWY	Quantum dot-based probes	In vitro studies	[219]
YLFFVFER	Quantum dots	In vivo and ex vivo studies	[221]
KLRLEWNR	Quantum dots	In vivo and ex vivo studies	[221]

## Data Availability

Not applicable.

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
