# Peer review of "Peptides Targeting HER2-Positive Breast Cancer Cells and Applications in Tumor Imaging and Delivery of Chemotherapeutics"

_nanomaterials, 2023, doi:10.3390/nano13172476_

Round 1
Reviewer 1 Report
The review is written comprehensively, detailed and understadable to nonspecialists. So why I do not have any remarks.
Author Response
We thank the reviewer for his/her positive evaluation.
Reviewer 2 Report
In this well-written review article, the authors systematically summarized the peptides that can target HER2-positive breast cancer cells and their wild applications in tumor imaging and targeted chemotherapeutics. This paper is of great interest for the researchers in nanomedicine and breast cancer treatment. The reviewer suggested the publication of this paper after minor revisions. The figure legends and captions are relatively brief. The authors should provided more detailed descriptions in the legends for readers' clear understandings. For example, in Figs. 2 and 3, the full names of the abbreviations and structures of different colors should be explained.
Author Response
We thank the reviewer for his/her positive comments and right remark. As suggested, we implemented the information (highlighted in yellow) in the legends of figures 2, 3, 4, 5, 6, 8, and 9.
Reviewer 3 Report
This review paper provides a comprehensive overview of peptides targeting HER2-positive breast cancer cells. It discusses the role of these peptides in tumor imaging and the delivery of chemotherapeutics. The paper is well-structured and provides a detailed analysis of the current state of research in this field. The paper is highly relevant to the field of oncology, particularly in the context of HER2-positive breast cancer. It provides a unique perspective on the use of peptides for targeted therapy, which is a rapidly evolving area of research. The paper is original in its approach and offers valuable insights into potential therapeutic strategies.
-Potential Pitfalls: The paper could have benefited from a more detailed discussion of the potential limitations and challenges associated with peptide-based therapies. While the authors do touch upon these issues, a more in-depth exploration would have added to the paper's comprehensiveness.
- Please add following 2 recently published articles: Bartsch R, Berghoff AS, Furtner J, Marhold M, Bergen ES, Roider-Schur S, Starzer AM, Forstner H, Rottenmanner B, Dieckmann K, Bago-Horvath Z, Haslacher H, Widhalm G, Ilhan-Mutlu A, Minichsdorfer C, Fuereder T, Szekeres T, Oehler L, Gruenberger B, Singer CF, Weltermann A, Puhr R, Preusser M. Trastuzumab deruxtecan in HER2-positive breast cancer with brain metastases: a single-arm, phase 2 trial. Nat Med. 2022;28(9):1840-1847.
Miglietta F, Griguolo G, Bottosso M, Giarratano T, Lo Mele M, Fassan M, Cacciatore M, Genovesi E, De Bartolo D, Vernaci G, Amato O, Porra F, Conte P, Guarneri V, Dieci MV. HER2-low-positive breast cancer: evolution from primary tumor to residual disease after neoadjuvant treatment. NPJ Breast Cancer. 2022;8:66.
Author Response
In accordance with the valuable suggestion of the referee, we have incorporated a detailed discussion (marked in yellow) concerning the potential limitations and challenges associated with peptide based therapies, at the end of the paragraph “3.2. HER2 receptor targeting peptides” This discussion also highlights the most pertinent strategies for overcoming these challenges. Consequently, we have suitably expanded the "References" section to include the relevant sources corres ponding to this enriched discussion (references 101-117).
Within the paragraph “3. HER2 positive breast cancer targeted therapies” therapies”, we have reported the two recommended articles: from line 220 to line 229 for " Nat Med. 2022;28(9):1840-1847", and from line 265 to line 277 for " NPJ Breast Cancer. 2022; 8:66 ". Subsequently, we have included the corresponding references in the designated "References" section.
Reviewer 4 Report
Review report
In the review titled: “Peptides targeting HER2-positive breast cancer cells and appli- 2
cations in tumor imaging and delivery of chemotherapeutics” is aimed to overview the main approaches for the diagnosis and treatment of HER2-positive breast cancer and is focused on the different targeting strategies involving antibodies and peptides. The review is paying attention on the role of nanotechnology for overcoming some of the current clinical challenges by using different types of HER2-targeting strategies by developing novel HER2-guided nanosystems which can be suitable for breast cancer imaging, targeting and therapy.
The present review is timely and well organized covering the most recent research on HER2 as a therapeutic target in breast cancer treatment. The review covers many aspects of main targeting strategies involving antibodies and peptides and using nanothehnology approach for the development of HER2-targeted nanosystems, which could address current clinical challenges and advance breast cancer imaging, targeting, and therapy.
The included tables and diagrams of excellent quality contribute to the comprehensibility of the review.
Author Response
We greatly appreciate the reviewer’s meticulous evaluation and positive feedback on our manuscript.